# Towards Universal & Efficient Model Compression via Exponential Torque Pruning

## Abstract

The rapid growth in complexity and size of modern deep neural networks (DNNs) has increased challenges related to computational costs and memory usage, spurring a growing interest in efficient model compression techniques. Previous state-of-the-art approach proposes using a Torque-inspired regularization which forces the weights of neural modules around a selected pivot point. Whereas, we observe that the pruning effect of this approach is far from perfect, as the post-trained network is still dense and also suffers from high accuracy drop. In this work, we attribute such ineffectiveness to the default linear force application scheme, which imposes inappropriate force on neural module of different distances. To efficiently prune the redundant and distant modules while retaining those that are close and necessary for effective inference, in this work, we propose Exponential Torque Pruning (ETP), which adopts an exponential force application scheme for regularization. Experimental results on a broad range of domains demonstrate that, despite its simplicity and ease of implementation, ETP manages to achieve significantly higher compression rate than the previous state-of-the-art pruning strategies with negligible accuracy drop.

## 1 Introduction

Deep neural networks (DNNs) have revolutionized countless domains by setting state-of-the-art baselines that significantly surpass previous approaches. However, nowadays DNNs are pretty large in size and require substantial floating point operations per second (FLOPS) for inference, which limits their applications in resource-constrained scenarios (e.g., edge devices Qin et al. (2018); Han et al. (2015b); Hinton (2015)). To achieve more efficient while also effective inference, many model compression techniques have been proposed. *E.g.*, low rank approximation, which aims to leverage a lower-rank matrix to capture the essential structure of the original model's weight matrix while reducing complexity Hu et al. (2022); Tiwary et al. (2025); Zanella and Ben Ayed (2024); un-

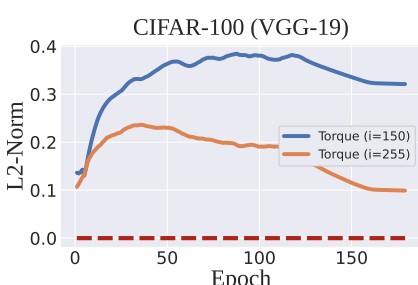

Figure 1: L2-norm curve during training process of VGG-19 on CIFAR-100.

structured pruning, which removes unimportant weights from DNNs to reduce its size while preserving performance LeCun et al. (1989); Liao et al. (2023); Muralidharan (2023); Structured pruning, which removes entire groups of weights (such as channels, filters, or blocks) from DNNs in a regular pattern, to reduce its size and computational cost while preserving performance Ding et al. (2018); He et al. (2018b); Fang et al. (2023b). Specifically, the previous state-of-the-art structured pruning method proposes a Torque-inspired regularization loss Gupta et al. (2024b), named as *Torque*. Concisely, analogous to the physical definition of "Torque", this approach uses a regularization that functions as a force that consolidates the weights of a neural module around a selected pivot point during training. By regularizing the model in this way, the weights of the neural modules that are far away from the pivot point would be forced to be zero, which could therefore be pruned. However, we observe that the vanilla Torque structured pruning is still far from perfect. Figure 1 shows the L2-norm curves of two neural modules during the training process. Concretely, the two neural modules are of different distances $d$ from a selected pivot module, which are all from the same

network layer[1]. It is obvious that though the two investigated modules are far from the pivot point (i.e., $d = 150$, $d = 255$), their L2-norm is quite high (i.e., $||\mathbf{w}|| \gg 0$, where $\mathbf{w}$ denotes the weight tensor), which is far from optimal. Thus, because of the low sparsity of the regularized model, it still suffers from a high accuracy drop after pruning, specifically, a 7% absolute accuracy reduction despite achieving only a $9\times$ speed-up[2] on the CIFAR-100 dataset for the VGG19 model.

In this paper, we attribute the ineffectiveness of Torque to its improper force application to modules of different distances. Concisely, Torque adopts a simple linear force application scheme, which applies inadequate force on distant neural modules while imposing unnecessarily large penalties on neural modules that are close to the pivot point, which are indispensable for effective inference. To mitigate this drawback, in this work, we propose Exponential Torque Pruning (ETP), which adopts an exponential force application scheme for regularization. By applying ETP, we could efficiently prune the redundant and distant modules by applying exponentially large forces that constrain the neural modules' weights to zero while retaining those that are close and necessary for effective inference.

Though being extremely simple and straightforward, we observe that ETP manages to achieve fascinating improvements over the previous state-of-the-art baselines on various domains. Besides, ETP is universal and is directly applicable to different model architectures. For example, on the natural language understanding tasks, ETP achieves a $42\times$ speed-up with merely 2.4% accuracy drop on BERT (SST-2), while the previous state-of-the-art (SoTA) pruning methods sustains $\geq 5\%$ accuracy drop; and on the image classification tasks, ETP achieves a $23\times$ speed-up with merely 4.5% drop in accuracy on VGG-19 (CIFAR-100), while the previous SoTA suffers from a large accuracy drop of 10.8%.

The contributions of our paper are as follows:

- We propose a universal structured pruning strategy, called Exponential Torque Pruning (ETP)[3], which leverages an exponential force application scheme that imposes a larger force on distant neural modules so as to constrain their weights to zero while preserving those that are close to the pivot point that are indispensable for effective inference.
- Experimental results on four distinct downstream domains and various model architectures validate that ETP can surpass the previous state-of-the-art pruning techniques regarding compression rate by a large margin, while retaining negligible accuracy drop.
- The significant improvement, high generality, and low additional training overhead pave the way for its strong potential in compressing modern Large Language Models (LLMs).

## 2 PRELIMINARY

In this section, we introduce the fundamentals of structured pruning and a state-of-the-art regularization-based structured pruning technique: torque-based structured pruning.

### 2.1 REGULARIZATION-BASED STRUCTURED PRUNING

To better understand regularization-based structured pruning, we first formally define the notion of neural module, which is as follows:

**Definition 1** (**Neural Module**). *A neural module is defined as a discrete, fundamental computational unit within a neural network that performs a specific transformation or operation on its input. Formally, given a neural network $\mathcal{N}$ parameterized by $\theta$, a neural module $M_i$ corresponds to a subset of parameters $\theta_i \subseteq \theta$ and a corresponding functional mapping*

$$M_i : \mathcal{X}_i \to \mathcal{Y}_i,$$

*where $\mathcal{X}_i$ and $\mathcal{Y}_i$ denote the input and output spaces of the module, respectively.*

---

[1]The first neural module in a layer is selected to be the pivot point, and the distance of a neural module from the pivot point is number of neural modules in between

[2]*speed-up* is a metric that measures the reduction in computational cost, which calculates the ratio of the total operations required in the baseline model to that in the pruned model.

[3]The implementation is available at: https://anonymous.4open.science/r/ETP-3EB6

Examples of neural modules include individual convolutional filters in a convolutional neural network (CNN) Ding et al. (2018); You et al. (2019), attention heads in a Transformer Ma et al. (2023); Fang et al. (2023b), or neurons within a fully connected layer Fang et al. (2023b); Gupta et al. (2024b). Regularization-based structured pruning is a vital technique in deep neural network compression Liu et al. (2017); Wang et al. (2020); Fang et al. (2023b); Gupta et al. (2024b); He and Xiao (2023b); Ding et al. (2019b); Fang et al. (2023a) which operates based on fundamental neural modules, where entire components such as filters, neurons, or layers are removed instead of individual weights. This leads to more efficient models that are computationally and memory efficient, making them ideal for deployment on resource-constrained devices. Given a network $\mathcal{N}(\mathbf{x}; \theta)$, where $\mathbf{x}$ denotes the input data, $\theta$ is the model's parameters, structured pruning aims to find a reduced set of parameters $\theta^* \subset \theta$ such that: $\mathcal{N}(\mathbf{x}; \theta^*) \approx \mathcal{N}(\mathbf{x}; \theta)$, while minimizing the network size. The pruning process is typically guided by a combined loss function:

$$\mathcal{L}_{\text{total}}(\theta^*) = \mathcal{L}_{\text{task}}(\mathcal{N}(\mathbf{x}; \theta^*)) + \lambda \mathcal{L}_{\text{pruning}}(\theta^*) \tag{1}$$

where $\mathcal{L}_{\text{task}}$ represents task-specific loss (e.g., classification), and $\mathcal{L}_{\text{pruning}}$ regularizes sparsity. Common types of structured pruning include filter pruning He et al. (2019); Li et al. (2016); Ding et al. (2019a; 2018), neuron pruning LeCun et al. (1989); Zhuang et al. (2020); Yu et al. (2018); Lee et al. (2019), channel pruning Gao et al. (2021); Wang et al. (2019); Ding et al. (2021); He et al. (2017), and layer pruning Fan et al. (2019); Wang et al. (2018b); Dong et al. (2017); Elkerdawy et al. (2020). These methods present unique challenges in balancing the trade-off between reduced size and maintaining task performance, with each approach requiring careful optimization to avoid excessive accuracy degradation.

## 2.2 TORQUE-BASED STRUCTURED PRUNING

Previous state-of-the-art pruning techniques require modifications to the network architecture or implementation of complex gradient update rules. Whereas, Gupta et al. Gupta et al. (2024a) propose a simple yet effective Torque-inspired approach (denoted as Torque in the following paper) which manages to achieve a great compression rate while requiring no change to model architecture. Analogous to the very definition of the physical concept (i.e., Torque), this approach proposes to apply a force to neural modules in order to consolidate the weights of a network layer around a selected pivot point during training. Formally, Torque approximates the concept with the following implementation:

$$||\tau_i^l||_2 = ||\mathbf{F}_i^l \times \mathbf{r}_i^l||_2 \approx ||\mathbf{w}_i^l||_2 \cdot d_i^l, \ i \in \mathbb{Z}^+ \tag{2}$$

where $\tau_i$ denotes the torque applied to the $i^{th}$ neural module of a layer $l$, $\mathbf{r}_i$ is the corresponding position vector. Torque-pruning approximates the L2-norm of the Torque that applies to the neural module's weights as the multiplication of the L2-norm of the module's weight matrix (i.e., force) $||\mathbf{w}_i^l||_2$ and the Euclidean distance of their corresponding indices $d_i^l = ||\rho_i^l - \rho_p^l||_2$, where $\rho_i^l$ is the index of the $i^{th}$ neural module, $\rho_p^l$ is the index of the pivot point. Gupta et al. (2024a) adopt a random indexing strategy for the modules within a layer, which performs well empirically. Given the Torque $\tau_i$, Torque-pruning proposes using it as a force that pushes the weights of neural modules that are distant from the pivot point to zero. Concretely, it implements it as a regularization term $\mathcal{L}_{\text{Torque}}$. The detailed optimization objective is as follows:

$$\mathcal{L}_{\text{total}}(\theta^*) = \mathcal{L}_{\text{task}}(\mathcal{N}(\mathbf{x}; \theta^*)) + \lambda \mathcal{L}_{\text{Torque}}(\theta^*) \tag{3}$$

$$= \mathcal{L}_{\text{task}}(\mathcal{N}(\mathbf{x}; \theta^*)) + \lambda \sum_l \sum_i ||\tau_i^l||_2 \tag{4}$$

where $\lambda$ denotes the regularization coefficient of $\mathcal{L}_{\text{Torque}}$. Figure 2(a) shows an intuitive visualization of the vanilla Torque regularization. Specifically, the $\mathcal{L}_{\text{Torque}}$ regularization imposes a penalty on neural modules proportional to their distance from the pivot point (i.e., $\frac{\partial \tau}{\partial ||\mathbf{w}||} \propto ||\rho_i^l - \rho_p^l||_2$); the further it is, the more its weights would be penalized (illustrated via the depth of the color of the representing circles).

## 3 EXPONENTIAL TORQUE PRUNING

In this section, we introduce the detailed motivation and design of our proposed method, called ETP (Exponential Torque Pruning). The key motivation of ETP is the sub-optimal force application

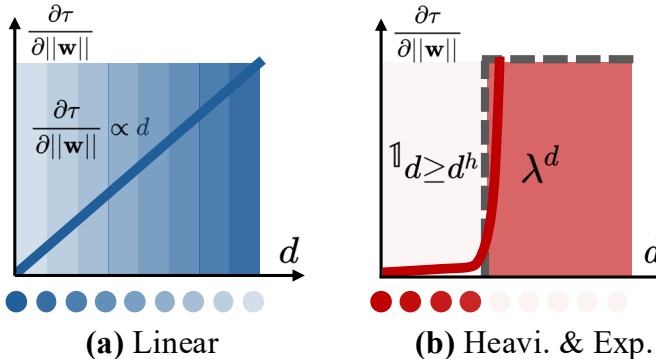

**(a)** Linear          **(b)** Heavi. & Exp.

Figure 2: (a) Visualization of vanilla Torque-prune regularization. The circles below the coordinate system, arranged from left to right, represent neural modules at corresponding distances from the pivot point (leftmost circle). The depth of the circle's fill color represents the L2-norm of the module. The lighter the color, the smaller the weight. The colored area within the coordinate denotes the magnitude of force applied to neural modules of different distances. The darker the color is, the greater the magnitude of the applied force is. (b) The ideal Heaviside Torque regularization and the exponential approximation.

scheme of the previous state-of-the-art Torque-prune approach. Specifically, we argue that the linear proportionality between the partial derivative of torque regularization with respect to the neural module's weights and the distance (i.e., $\frac{\partial \tau}{\partial ||\mathbf{w}||} \propto ||\rho_i^l - \rho_p^l||_2$) is inappropriate. Intuitively, this denotes that for neural modules of different distances from the pivot point, we are applying the same amount of force to drive them to zero. As demonstrated in Figure 1, such linear force application scheme fails to constrain the weights of modules that are distant from the pivot point, while inappropriately penalizing the ones that are close and necessary for inference. To achieve a sparser and effective network architecture, we propose using a nonlinear force application scheme. In particular, we should apply a much larger force on the distant neural modules to drive them towards zero, while a smaller or no force (i.e., penalty) on those that are close to the pivot point, since they are essential for effective inference. Figure 2(a) and Figure 2(b) show the intuitive visualization of different force application schemes as well as the corresponding illustration of the regularized neural modules' L2-norm (denoted as circles filled with colors below the distance axis). The colored area within the coordinate denotes the magnitude of force applied to neural modules of different distance. The darker the color is, the greater the magnitude of the applied force is.

Concretely, we can observe from the colored blue area within the coordinate in Figure 2(a) that due to the linear force application scheme of the vanilla Torque-prune, all the neural modules are penalized in a linear manner according to their distant from the pivot point (denoted as the origin). This results in an undesirable weight distribution (shown as the blue circles sequence) where neural modules farther from the pivot point remain densely weighted, while the neural modules that are close to the pivot point (which are considered necessary for effective inference) are inappropriately penalized. Ideally, the force application scheme should exert a large penalty on the neural modules that are distant from the pivot point while fully preserving the neural modules that are close to the pivot point and are necessary for effective inference. Therefore, we formulate such nonlinear force application scheme using a Heaviside step function (denoted as the dashed gray line) in Figure 2(b), such that $\frac{\partial \tau}{\partial ||\mathbf{w}||} \propto \mathbb{1}_{||\rho_i^l - \rho_p^l||_2 \geq d^h}$, where $d_h$ denotes the threshold distance from the pivot point, $\mathbb{1}_{||\rho_i^l - \rho_p^l||_2 \geq d^h}$ denotes the indicator function that outputs 1 if the relative distance of the investigated neural module and the pivot point $||\rho_i^l - \rho_p^l||_2$ is larger than $d_h$, otherwise it outputs 0. Thus, the detailed implementation of the Heaviside Torque regularization $||\tilde{\tau}_i^l||_2$ is as follows:

$$||\tilde{\tau}_i^l||_2 = ||\mathbf{F}_i^l \times \mathbf{r}_i^l||_2 \tag{5}$$

$$\approx ||\mathbf{w}_i^l||_2 \cdot (\epsilon \cdot \mathbb{1}_{||\rho_i^l - \rho_p^l||_2 \geq d^h}), \ i \in \mathbb{Z}^+ \tag{6}$$

Specifically, for neural modules within the threshold distance $d_h$, we apply zero force on them since they are considered necessary for inference, and for the modules that are beyond $d_h$, we exert a large

force which is of size $\epsilon$ on these modules, driving their weights toward zero. Whereas the Heaviside step function is non-differentiable, we therefore use an exponential function for approximation (denoted as the solid red line in Figure 2(b)), formally:

$$||\hat{\tau}_i^l||_2 = ||\mathbf{F}_i^l \times \mathbf{r}_i^l||_2 \tag{7}$$

$$\approx ||\mathbf{w}_i^l||_2 \cdot \lambda^{||\rho_i^l - \rho_p^l||_2}, \ i \in \mathbb{Z}^+ \tag{8}$$

where $\lambda$ is a hyperparameter that serves as the base of the exponentiation that controls the threshold distance. Finally, given the exponential approximation of the Heaviside Torque regularization, the overall optimization objective with the exponential torque pruning regularization (ETP) loss is:

$$\hat{\mathcal{L}}_{\text{total}}(\mathbf{w}) = \mathcal{L}_{\text{task}}(\mathbf{x}; \mathbf{w}) + \beta \cdot \mathcal{L}_{\text{ETP}}^{\mathbf{w}} \tag{9}$$

$$= \mathcal{L}_{\text{task}}(\mathbf{x}; \mathbf{w}) + \beta \sum_l \sum_i ||\mathbf{w}_i^l||_2 \cdot \lambda^{||\rho_i^l - \rho_p^l||_2} \tag{10}$$

where $\mathcal{L}_{\text{task}}$ is the original optimization objective of the specific task, $\beta$ is the regulatory coefficient of the ETP loss. Therefore, by exerting extremely low penalty on the neural modules that are close to pivot point and are considered essential for effective inference, while much larger penalty on those that are distant from the pivot point, the exponential force application scheme introduced by ETP are expected to achieve a much sparser yet effective model architecture.

## 4 EXPERIMENTS

To evaluate the effectiveness of ETP, we conduct experiments on four distinct domains, including vision, language, graph, and time series. In the remainder of this section, we first introduce the detailed experimental setup, and then we answer three research questions (RQs) to lead our discussion, which are as follows:

**(RQ1) Speed-up Improvement** How effective is ETP in improving the models' speed-up while retaining a low performance drop?

**(RQ2) Aggresive Pruning Analysis** Does ETP perform well under different speed-up ratios?

**(RQ3) Effectiveness on Large Models** Can ETP effectively compress the prevalent large models?

### 4.1 EXPERIMENTAL SETUP

We demonstrate the effectiveness of ETP by evaluating it on multiple benchmarks of different domains. The details of the benchmarks are as follows[4]:

**Datasets & Backbone Models.** For the image classification tasks, we evaluate on CIFAR-100 Krizhevsky et al. (2009), and ImageNet Deng et al. (2009) datasets. CIFAR-100 consist of 50,000 training and 10,000 test images of size $32 \times 32$, with 100 classes respectively. All images are normalized using dataset-specific RGB means and standard deviations and resized to $224 \times 224$. For ImageNet, we use the ILSVRC-2012 subset with 1.2 million training images and 50,000 validation images across 1,000 classes. The images are resized and randomly cropped to $224 \times 224$ during training. In terms of the backbone models, we follow the setup of Gupta et al. Gupta et al. (2024a) and conduct experiments on Vision Transformers (i.e., ViT-B/16 Dosovitskiy et al. (2020)), CNN models with linear connections only (i.e., VGG-19 Simonyan and Zisserman (2014)), and the ones with residual connections (ResNet-50 He et al. (2016)). For the graph classification task, we use the Protein-Protein Interaction (PPI) dataset Hamilton et al. (2017), which contains 24 graphs with over 56,000 nodes and 818,000 edges, each node is represented with 50-dimensional features. We adopt the Graph Attention Network (GAT) Veličković et al. (2017) as the backbone model. For the domain of natural language understanding (NLU), we evaluated two GLUE benchmark datasets: SST-2 and MRPC Wang et al. (2018a). SST-2 is a sentiment classification task with 67,349 training and 872 test examples, while MRPC is a paraphrase detection task with 3,668 training and 408 test pairs. We use BERT Devlin et al. (2019) and RoBERTa Liu et al. (2019) models as the backbone models for evaluation.

---

[4]Please refer to the appendix for the detailed experimental setup.

For the time-series forecasting task, we use the ETTh1 dataset from the ETT benchmark suite Zhou et al. (2021). The dataset includes hourly energy consumption features across one week, with 7 input features and 1 target variable. We adopt the Informer model Zhou et al. (2021) as the backbone, using an input sequence length of 96 and forecasting 48 future time steps. The features are normalized using the z-score normalization based on the training data statistics.

**Compared Methods.** We mainly compare ETP with the vanilla Torque pruning Gupta et al. (2024b) and DepGraph Fang et al. (2023b) Apart from these general-purpose baselines, we also compare ETP against the domain-specific SoTA baselines. Concretely, for the vision tasks, we also compare ETP against HRank Lin et al. (2020), SFP He et al. (2018a), and GReg Wang et al. (2020); For the natural language understanding (NLU) tasks, we further benchmark ETP against CoFi Xia et al. (2022), DynaBERT Hou et al. (2020), EBERT Liu et al. (2021), and LLM-Pruner Ma et al. (2023). Please refer to the Appendix for more detailed introduction of these methods.

Table 1: Pruning results on vision benchmarks. We highlight the top-1 results in red.

| CIFAR100 (VGG 19) | | | | |
|---|---|---|---|---|
| Method | Base | Pruned | Acc. Drop | Speed-up |
| GReg-1 | 74.02 | 67.35±0.15 | -6.67±0.15 | 8.84× |
| GReg-2 | 74.02 | 67.75±0.18 | -6.27±0.18 | 8.84× |
| Depgraph | 73.50 | 70.39±0.04 | -3.11±0.04 | 8.92× |
| Torque (r) | 73.03 | 65.87±0.21 | -7.16±0.21 | 8.88× |
| **ETP (Ours)** | 73.50 | **71.30±0.08** | **-2.20±0.08** | 9.03× |
| ImageNet (ResNet50) | | | | |
| Method | Base | Pruned | Acc. Drop | Speed-up |
| HRank | 76.15 | 74.98±0.46 | -1.17±0.46 | 1.78× |
| SFP | 76.15 | 74.51±0.32 | -1.64±0.32 | 1.72× |
| GReg-2 | 76.13 | 75.16±0.12 | -0.97±0.12 | 1.49× |
| Depgraph | 76.15 | 75.53±0.28 | -0.62±0.28 | 2.08× |
| Torque (p) | 76.07 | 74.67±0.11 | -1.40±0.11 | 2.34× |
| **ETP (Ours)** | 76.15 | **76.21±0.01** | **+0.06±0.01** | 2.30× |
| ImageNet (ViT-B/16) | | | | |
| Method | Base | Pruned | Acc. Drop | Speed-up |
| CP-ViT | 81.07 | 77.36±0.22 | -3.71±0.22 | 1.69× |
| DepGraph + EMA | 81.07 | 79.58±0.47 | -1.39±0.47 | 1.69× |
| DepGraph | 81.07 | 79.17±0.21 | -1.9±0.21 | 1.69× |
| **ETP (Ours)** | 81.07 | **81.93±0.51** | **+0.86±0.51** | 1.69× |

**Evaluation Measurements.** An ideal model compression algorithm should control the compression-accuracy tradeoff well, i.e., that is to effectively reduce the models' size, therefore reducing the number of computations for inference while controlling the accuracy loss within an acceptable range. To evaluate the compression-accuracy tradeoff quantitatively, we follow previous literature Fang et al. (2023b); Gupta et al. (2024b); Wang et al. (2020) and adopt the `speed-up` and `accuracy-drop` metrics for evaluation. Specifically, speedup is defined as follows: $\texttt{speed-up} = \frac{\text{MACS}_{\text{base}}}{\text{MACS}_{\text{pruned}}}$, MACS (Multiply-Accumulate operations) denotes the total number of arithmetic operations required for a single forward pass of the model. This is often used to approximate computational cost and inference latency. Intuitively, `speed-up` quantifies how much more efficient the pruned model is compared to the original model. A higher value indicates greater computational savings, which enables faster inference and lower energy consumption. The accuracy drop is defined as: $\texttt{accuracy-drop} = \text{accuracy}_{\text{pruned}} - \text{accuracy}_{\text{base}}$ Concretely, the `accuracy-drop` measures the accuracy loss of the model before and after pruning. Other task-specific metrics are further elaborated in the Appendix. Note that all the reported results are averaged over five random seeds.

### 4.2 SPEED-UP IMPROVEMENT (RQ1)

To answer RQ1, we compare ETP against the state-of-the-art pruning techniques on four domains (i.e., image classification, natural language understanding (NLU), graph classification, and time-series forecasting). The detailed results are illustrated in Table 1, Table 2, Table 4, and Table 3 respectively. First, for image classification tasks, we compare ETP with the state-of-the-art pruning baselines on ImageNet on ResNet-50 and ViT-B/16, and CIFAR-100 with VGG-19 model, following the same experimental setup as previous literature Fang et al. (2023b); Gupta et al. (2024b). The results in Table 1 show that ETP performs consistently better than the previous state-of-the-art baselines on all the investigated backbone models. Specifically, on the CIFAR-100 dataset using the VGG-19 backbone model, ETP achieves a 9× speed-up while incurring only a 2.2% drop in classification accuracy, while Torque and DepGraph suffers from a significantly higher accuracy degradation of 7.16% and 3.11% respectively. Moreover, ETP attains a 1.69× speed-up on ViT-B/16 without any loss in accuracy, whereas competing approaches typically suffer a 1.5–2% accuracy reduction under comparable speed-up.

For the NLU, graph classification, and time-series forecasting tasks, we observe that ETP generally surpasses both the general-purpose structured pruning methods as well as the state-of-the-art task-specific baselines as well. Specifically, for the GAT on the PPI dataset, we

Table 2: Pruning results on the NLU benchmarks.

| | SST-2 | | | | | | | |
|---|---|---|---|---|---|---|---|---|
| **Method** | **BERT** | | | | **RoBERTa** | | | |
| | **Base** | **Pruned** | **Acc. Drop** | **Speed-up** | **Base** | **Pruned** | **Acc. Drop** | **Speed-up** |
| CoFi | 93.5% | 87.6% | -5.9% | 11× | 95.3% | 80.0% | -15.3% | 13.5× |
| DynaBERT | 93.5% | 85.1% | -8.4% | 11× | 95.3% | 78.1% | -17.2% | 13.5× |
| EBERT | 93.5% | 86.0% | -7.5% | 11× | 95.3% | 78.7% | -16.6% | 13.5× |
| DepGraph | 93.5% | 91.8% | -1.7% | 11× | 95.3% | 89.9% | -5.4% | 13.5× |
| LLM-Pruner | 93.5% | 91.8% | -1.7% | 11× | 95.3% | 90.3% | -5% | 13.5× |
| Torque | 93.5% | 90.9% | -2.6% | 11× | 95.3% | 90.6% | -4.7% | 13.5× |
| **ETP (Ours)** | 93.5% | **92.1%** | **-1.4%** | 11× | 95.3% | **92.9%** | **-2.4%** | 13.5× |
| | MRPC | | | | | | | |
| **Method** | **BERT** | | | | **RoBERTa** | | | |
| | **Base** | **Pruned** | **Acc. Drop** | **Speed-up** | **Base** | **Pruned** | **Acc. Drop** | **Speed-up** |
| CoFi | 88.0% | 80.3% | -7.7% | 8× | 90.0% | 82.4% | -17.6% | 8× |
| DynaBERT | 88.0% | 79.6% | -8.4% | 8× | 90.0% | 83.8% | -16.2% | 8× |
| EBERT | 88.0% | 74.5% | -13.5% | 8× | 90.0% | 81.1% | -18.9% | 8× |
| DepGraph | 88.0% | 83.5% | -4.5% | 8× | 90.0% | 86.1% | -3.9% | 8× |
| LLM-Pruner | 88.0% | 83.0% | -5% | 8× | 90.0% | 85.9% | -4.1% | 8× |
| Torque | 88.0% | 83.2% | -4.8% | 8× | 90.0% | 85.3% | -4.7% | 8× |
| **ETP (Ours)** | 88.0% | **85.0%** | **-3.0%** | 8× | 90.0% | **86.6%** | **-3.4%** | 8× |

evaluate different methods under two speed-up settings. For speed-up=12×, ETP achieves a F1 score drop of only 0.027, while DepGraph incurs a F1 score drop of 0.03 for the same speed-up. The results on 9× speed-up rate are similar. Despite our best efforts, the vanilla Torque method fails to achieve the 12× speed-up on the GAT (PPI).

We believe that this is because Torque is unable to penalize the GAT models' parameters enough to make them structurally sparse.

ETP outperforms DepGraph and Torque on Informer for Etth-1 dataset as well for speed-up $\geq 6.5\times$ for both MAE and MSE. For speed-up $\leq 4\times$, ETP consistently outperforms DepGraph with a significant performance gain and is competent or slightly worse than Torque. The results indicate that ETP's superiority is more sig-

Table 3: Pruning results on the Informer model.

| | Etth-1(48) (Informer) | | | | | |
|---|---|---|---|---|---|---|
| | DepGraph | | Torque | | **ETP (Ours)** | |
| Speed-Up | MAE | MSE | MAE | MSE | MAE | MSE |
| 1× | 0.319 | 0.158 | 0.319 | 0.158 | 0.319 | 0.158 |
| 2.5× | 0.3559 | 0.1636 | **0.3398** | 0.1621 | 0.3402 | **0.1618** |
| 4× | 0.3632 | 0.1671 | **0.3492** | 0.1665 | 0.3495 | **0.1631** |
| 6.5× | 0.3737 | 0.1702 | 0.3606 | 0.1698 | **0.3580** | **0.1645** |
| 10.5× | 0.3818 | 0.1743 | 0.3723 | 0.1756 | **0.3643** | **0.1661** |
| 14.5× | 0.3959 | 0.1797 | 0.3843 | 0.1810 | **0.3726** | **0.1678** |
| 25× | 0.4118 | 0.1852 | 0.3937 | 0.1843 | **0.3812** | **0.1692** |

nificant under large speed-up rate scenarios. The reason is that with more redundant parameters (i.e. more distant and redundant neural modules) awaiting to be pruned, ETP can achieve more effective pruning by applying much larger penalty on the redundancy while retaining the modules that are necessary for inference according to its exponential force application scheme .

To have a better understanding of the improvement, we also present wall-clock latency and energy consumption results across 5 hardware platforms: NVIDIA A100, L4, RTX 8000, Tesla T4, and Google TPU v6. The results are shown in Table 5. We present the results of BERT on SST-2 and VGG-19 on CIFAR-100. Specifically, ETP achieves 6.4–9.3× latency speed-up on GPUs and TPUs. It

Table 4: Pruning results on the GAT model.

| | PPI (GAT) | | | |
|---|---|---|---|---|
| **Method** | **Base** | **Pruned** | **F1 score Drop** | **Speed-Up** |
| DepGraph | 0.9860 | 0.9610±0.0000 | -0.0250±0.0000 | 8.43× |
| Torque | 0.9860 | - | - | - |
| **ETP (Ours)** | 0.9860 | **0.9701±0.0010** | **-0.0159±0.0010** | 9.13× |
| DepGraph | 0.9860 | 0.9555±0.0007 | -0.0345±0.0007 | 12× |
| Torque | 0.9860 | - | - | - |
| **ETP (Ours)** | 0.9860 | **0.9624±0.0005** | **-0.0236±0.0005** | 12.16× |

also attains a 56–90% energy reduction depending on device class.

To better understand the source of improvement, we conduct an in-depth analysis to track the progress of the L2-norm of specific neural modules during the learning process. Concretely, we randomly select two neural modules within a specific layer that are of different distances from the pivot point, we compare the L2-norm learning process of ETP and the vanilla Torque pruning approach. The results are shown in Figure 3. We can observe that compared with the vanilla Torque, ETP can significantly reduce the L2-norm of the distant neural modules, e.g., for VGG-19 trained on CIFAR-100, ETP manages to optimally prune the distant module (i.e., L2-norm equals to 0 ($\|m_{254}^l\| = 0.0$)), while Torque remains a high L2-norm (i.e., $\|m_{254}^l\| = 0.134$). The extensive L2-norm analysis during training validates that the exponential force application scheme enables ETP to achieve a significantly

Table 5: Cross-hardware evaluation of latency and energy for pruned BERT (SST-2) and VGG-19 (CIFAR-100). Theoretical speed-ups are $11\times$ (BERT) and $9\times$ (VGG-19).

| Model | Hardware | Base Lat. (ms) | Pruned Lat. (ms) | Speedup | Base Energy (J) | Pruned Energy (J) | Reduction |
|---|---|---|---|---|---|---|---|
| BERT SST-2 (92.1%) | NVIDIA A100 | 45.863±0.115 | 7.098±0.377 | 6.46× | 14.471±0.605 | 2.439±0.722 | 83.1% |
| | NVIDIA L4 | 101.166±1.258 | 10.868±0.341 | 9.31× | 7.492±0.082 | 0.787±0.019 | 89.5% |
| | Quadro RTX 8000 | 66.540±0.608 | 9.650±0.044 | 6.90× | 17.547±0.173 | 3.507±0.551 | 80.0% |
| | NVIDIA Tesla T4 | 188.705±3.687 | 23.293±0.486 | 8.10× | 14.585±0.586 | 1.622±0.079 | 88.9% |
| | Google TPU v6 | 546.013±12.463 | 79.637±8.126 | 6.86× | 73.390±4.653 | 11.277±3.972 | 84.6% |
| VGG-19 CIFAR-100 (71.30%) | NVIDIA A100 | 5.578±0.004 | 0.818±0.013 | 6.82× | 1.853±0.559 | 0.278±0.119 | 85.3% |
| | NVIDIA L4 | 14.914±0.151 | 6.117±0.084 | 4.43× | 1.080±0.010 | 0.240±0.000 | 77.4% |
| | Quadro RTX 8000 | 15.013±0.065 | 3.538±0.804 | 4.25× | 3.535±0.023 | 1.556±0.222 | 56.0% |
| | NVIDIA Tesla T4 | 35.426±0.192 | 9.262±0.116 | 3.82× | 2.521±0.273 | 1.090±0.230 | 56.7% |
| | Google TPU v6 | 171.202±6.831 | 19.641±0.373 | 8.72× | 22.934±2.0142 | 2.630±0.371 | 88.5% |

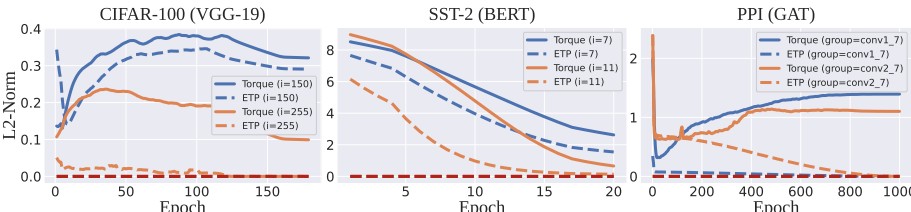

Figure 3: Comparison of the L2-norm curves during the training process.

sparser yet effective neural network architecture, resulting in a much higher compression rate with minimal performance degradation.

### 4.3 AGGRESSIVE PRUNING ANALYSIS (RQ2)

Different real-world applications require different levels of model compression due to hardware limitations; therefore, to perform well (i.e., retain low accuracy drop) under different speed-up ratios is a crucial ability for the model compression techniques. To systematically quantify such ability, we propose evaluating different pruning methods via the **aggressive pruning analysis**. Concretely, that is to record the model's accuracy drop across progressively increasing speed-up ratios. We conduct the analysis on the six different tasks, the results are shown in Figure 4. It is obvious that ETP manages to retain the accuracy within an acceptable range while the other compared methods suffer from a significant accuracy drop. For example, for BERT on MRPC, under the $30\times$ speed-up ratio, ETP achieves an accuracy drop of only 3.5%, while DepGraph and Torque's accuracy drop by 6% and 6.1% respectively. Similarly, for VGG19 on CIFAR 100 dataset, under the $23\times$ speed-up setting, ETP incurs an accuracy drop of only 3.87%, while DepGraph and GReg's accuracy is 10.73% and 13% respectively. We observe similar trends on Informer for ETTh-1 (48) dataset as well. ETP incurs a change in MSE of 0.02 for a $38\times$ speed-up, while DepGraph incurs a change in MSE of 0.032 for the same speed-up. Torque performs the worst out of the 3 methods at $38\times$ speed-up and incurs a change in MSE of 0.041. The results demonstrate that, thanks to a more reasonable force application scheme, ETP is a much more robust pruning technique and it is more suitable for scenarios that require a large speed-up ratio (e.g., model deployment on edge devices with limited computing power) compared to the previous state-of-the-art baselines.

### 4.4 EFFECTIVENESS ON LARGE MODELS (RQ3)

To demonstrate ETP's potential in compressing large models, we further evaluate ETP on the OPT-350M Zhang et al. (2022) language model using the WikiText Merity et al. (2016) dataset, with perplexity as the evaluation metric. All methods are constrained to 50% sparsity for a fair comparison. As summarized in Table 16, unstructured magnitude pruning fails under this budget (perplexity $6 \times 10^3$), while structured baselines such as SparseGPT Frantar and Alistarh (2023), Wanda Sun et al. (2023), and DepGraph Fang et al. (2023b) achieve perplexities in the 32–36 range. LLM-Pruner Ma et al. (2023), a recent method specialized for large language models, improves performance to 31.05. ETP achieves 29.14 perplexity, surpassing all baselines under identical sparsity. This corresponds to a 6–10% relative improvement over SparseGPT and DepGraph, which narrows the gap to the dense

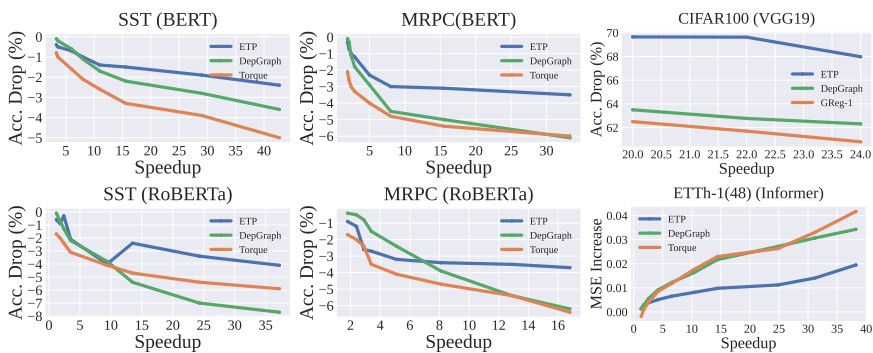

Figure 4: Results of aggressive pruning analysis for six distinct tasks.

baseline (22.00). These results highlight ETP's ability to retain model quality across autoregressive language modeling tasks, suggesting its potential as a general pruning framework for large models. We leave the comprehensive evaluation of ETP on other large language models to future work.

## 5  RELATED WORKS

**Unstructured pruning** aims to remove individual weights in a network, typically based on magnitude-based heuristics or importance scores LeCun et al. (1989); Muralidharan (2023); Dong et al. (2017); Lee et al. (2019). One of the seminal works in this area was introduced by Han et al. (2015a), who proposed an iterative pruning framework that eliminates weights with small magnitudes and then retrains the network to recover any lost accuracy. This approach was shown to significantly reduce model size while maintaining competitive performance.

**Structured pruning** focuses on removing higher-level structures, such as entire channels, filters, or even layers.

Table 6: Pruning results on OPT-350M [WikiText].

| WikiText (OPT350M) | | |
|---|---|---|
| **Method** | **Sparsity** | **Perplexity** |
| Dense | 50% | 22.00 |
| Magnitude | 50% | $6 \times 10^3$ |
| SparseGPT | 50% | 34.76 |
| Wanda | 50% | 35.92 |
| DepGraph | 50% | 32.61 |
| LLM-Pruner | 50% | 31.05 |
| **ETP (Ours)** | **50%** | **29.14** |

This yields a compact and dense model architecture that is more compatible with conventional hardware and software frameworks. He and Xiao (2023a); Ding et al. (2018); He et al. (2018b); Ding et al. (2021); You et al. (2019); Lin et al. (2020) Early approaches, such as that by Li et al. (2016), prune filters in convolution layers based on their $\ell_1$ norm, under the assumption that filters with smaller norms contribute less to the final output. He et al. (2017) proposed channel pruning guided by evaluating the change in loss when specific channels are removed, allowing for a more data-driven pruning strategy. These methods are typically followed by fine-tuning to restore the performance of the pruned network Lin et al. (2020); Fang et al. (2023b). Recent advances have cast structured pruning as a learning or optimization problem.

## 6  CONCLUSION

In this work, motivated by the observation that the vanilla Torque-based pruning still fails to achieve satisfying model sparsity, we introduce a simple yet effective pruning method called Exponential Torque Pruning (ETP) based on an exponential force application scheme. Concretely, ETP imposes a larger force on distant neural modules so as to constrain their weights to zero while preserving those that are close to the pivot point that are indispensable for effective inference. Experiments across four diverse downstream domains and multiple model architectures (including modern large language models) demonstrate that despite its simplicity and ease of implementation, ETP significantly outperforms prior state-of-the-art pruning methods, achieving a much higher compression rate while maintaining considerably lower accuracy degradation. Future progress could be made in the force application scheme to mitigate this limitation. Besides, we also plan to apply ETP to other emerging architectures (e.g., diffusion models, Mixture-of-Experts architectures, etc.) to further assess its generalizability.

**Reproducibility Statement:** Our implementation is publicly available at `https://anonymous. 4open.science/r/ETP-3EB6`, including training, evaluation, and preprocessing scripts. All datasets used in this work are publicly accessible, and preprocessing steps are followed in accordance with State-of-The-Art. Comprehensive details of model configurations, hyperparameters, and training schedules are provided in section 7.1. Experiments were conducted on a single NVIDIA A100 GPU.

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

# 7 APPENDIX

## 7.1 TRAINING SETUP

Table 7: Training configurations for ETP across all evaluated benchmarks. Standard schedules are used per task to ensure fair comparison with the compared baselines.

| Dataset (Model) | Epochs | Batch Size | LR Scheduler | Optimizer |
|---|---|---|---|---|
| CIFAR-10 (ResNet-56) | 100 | 128 | MultiStepLR (milestones = [60, 80], $\gamma = 0.1$) | SGD (lr = 0.001, momentum = 0.9, weight decay = 5e−4) |
| CIFAR-100 (VGG-19) | 100 | 128 | MultiStepLR (milestones = [60, 80], $\gamma = 0.1$) | SGD (lr = 0.001, momentum = 0.9, weight decay = 5e−4) |
| ImageNet-1k (ResNet-50) | 90 | 256 | StepLR (step size = 30, $\gamma = 0.1$) | SGD (lr = 0.1, momentum = 0.9, weight decay = 1e−4) |
| MRPC (BERT) | 10 | 32 | Linear decay with 10% warm-up | AdamW (lr = 2e−5, weight decay = 0.01) |
| SST-2 (BERT) | 10 | 32 | Linear decay with 10% warm-up | AdamW (lr = 2e−5, weight decay = 0.01) |
| MRPC (RoBERTa) | 10 | 32 | Linear decay with 10% warm-up | AdamW (lr = 2e−5, weight decay = 0.01) |
| SST-2 (RoBERTa) | 10 | 32 | Linear decay with 10% warm-up | AdamW (lr = 2e−5, weight decay = 0.01) |
| PPI (GAT) | 1000 | 1 | CosineAnnealingLR ($T_{\max} = 1000$) | Adam (lr = 0.005, weight decay = 5e−4) |
| ETTh1 (Informer) | 6 | 32 | CosineAnnealingLR ($T_{\max} = 6$) | Adam (lr = 5e−4, weight decay = 1e−4) |
| C4 (OPT-350M) | 5 | 64 | Linear warmup; LR ($1\times10^{-5}$) | AdamW (lr = 1e−5); gradient clipping = 1.0 |

In this section, we detail the training configuration of our proposed method and the baseline approaches. First, for ETP, we use the following strategy for all tasks to select the $\lambda$ and $\beta$ for our loss function. The regularization coefficient $\beta$ is selected via grid search over the range $\{10^{-6},\ 5 \times 10^{-6},\ 10^{-5},\ 5 \times 10^{-5},\ 10^{-4},\ 5 \times 10^{-4},\ 10^{-3}\}$. The optimal value of $\beta$ varies depending on the model architecture and the desired pruning aggressiveness. Higher compression rates are obtained by increasing $\beta$ accordingly. For the exponential base $\lambda$, we defined it as a function of the number of grouped parameters in a layer $l$: $\lambda_l = \exp\left(\frac{5}{|\mathcal{G}_l|}\right)$, where $|\mathcal{G}_l|$ denotes the total number of parameter groups (e.g., convolutional filter, attention head, etc.). The detailed training setup of ETP for all the evaluated benchmarks is illustrated in Table 7. We strictly follow the experimental setup of the compared baselines according to their provided implementations for fair comparison.

## 7.2 DETAILED INFORMATION ABOUT THE COMPARED METHODS

We compare all experiments against two general-purpose baselines namely, DepGraph Fang et al. (2023b) and Torque Gupta et al. (2024b):

1. **DepGraph**: DepGraph introduces a dependency-graph-based perspective for structured pruning in deep neural networks, where the pruning of one module (e.g., a convolutional filter or neuron) inherently affects the computational graph downstream. Mathematically, the network is represented as a graph $G = (V, E)$, where vertices $v \in V$ correspond to computational operators (e.g., filters, channels, or layers) and edges $e \in E$ represent data-flow dependencies. Pruning is then formulated as an optimization problem under dependency constraints:

$$\min_{\mathcal{M}} \ \mathcal{L}(f_{\mathcal{M}}(x), y) \quad \text{s.t.} \quad \mathcal{M} \subseteq V, \ \mathcal{M} \text{ satisfies dependency closure,}$$

where $\mathcal{M}$ denotes the set of retained modules, $\mathcal{L}$ the task loss, and dependency closure ensures that if a vertex is preserved, all of its prerequisite vertices along $G$ are also preserved.

2. **Torque**: The Torque Structured Pruning method introduces a physics-inspired regularization during training that encourages weight concentration near a chosen pivot filter while pushing

peripheral filters toward zero. The regularization loss is represented as:

$$\mathcal{L}_{\text{tot}} = \mathcal{L}_{\text{task}} + \lambda_T \sum_n \|T_n\|_2,$$

where

$$\|T_n\|_2 = \|W_n\|_2 \cdot |r_n - r_p|,$$

approximating the physical torque $F \times r$.

We also compare against domain-specific techniques. For vision tasks we compare against:

1. **HRank:** HRank introduces a filter pruning method that quantifies filter importance by the rank of their induced feature maps. Specifically, denote the feature map for filter $j$ in layer $i$ on input $I$ by $o_{ij}(I)$. HRank defines the importance score as

$$\hat{L}(o_{ij}) = \mathbb{E}_{I \sim P(I)}\big[\text{Rank}(o_{ij}(I))\big] \approx \frac{1}{g} \sum_{t=1}^{g} \text{Rank}\big(o_{ij}(I_t)\big),$$

where $\{I_t\}_{t=1}^{g}$ is a small sampled batch. Filters whose scores fall among the lowest in a layer are pruned.

2. **SFP**: Soft Filter Pruning (SFP) introduces a pruning scheme for CNNs in which filters are not permanently removed but instead set to zero and allowed to be updated during training. This differs from conventional hard pruning, where pruned filters are discarded and the network capacity is irreversibly reduced. At the end of each training epoch, a proportion of filters with the smallest $\ell_2$-norms are selected and reset:

$$W_j^{(l)} \leftarrow 0, \quad \text{for } j \in \mathcal{P}_l,$$

where $W_j^{(l)}$ denotes the $j$-th filter in layer $l$ and $\mathcal{P}_l$ is the set of pruned filters determined by norm ranking. Gradient descent continues to update all filters, including those zeroed, so that previously pruned filters may recover if they become useful.

3. **GReg**: It proposes a pruning framework in which a sparsity-inducing regularization term increases gradually during training, allowing the network to adapt smoothly to pruning pressure. Instead of applying a fixed strong regularizer from the start, GReg introduces a time-dependent weighting:

$$\mathcal{L}_{\text{tot}} = \mathcal{L}_{\text{task}} + \lambda(t)\, R(W),$$

where $R(W)$ denotes a structured sparsity regularizer (e.g. $\ell_{2,1}$ norm over filters or channels), and $\lambda(t)$ is a monotonically increasing function of the training step $t$.

For NLU tasks we compare against:

1. **CoFi:** This method introduces a unified *compression-and-fine-tuning* framework for pre-trained transformers, in which structured pruning and task adaptation are optimized jointly rather than sequentially. The method applies learnable binary masks $\mathbf{m}$ to weight groups at multiple granularities (attention heads, intermediate dimensions, hidden layers) and optimizes

$$\min_{\theta, \mathbf{m}} \ \mathcal{L}_{\text{task}}(f_{\theta \odot \mathbf{m}}(x), y) \ + \ \lambda\, \|\mathbf{m}\|_0,$$

where $\theta$ are pretrained parameters, $\mathbf{m}$ are structured pruning masks, and $\lambda$ controls sparsity. This method is primarily evaluated on BERT and RoBERTa.

2. **DynaBERT:** It proposes an adaptive width–depth pruning framework for transformers, training a single BERT that can dynamically adjust hidden dimensions (width) and number of layers (depth) to meet resource budgets. The method first conducts *width-adaptive training*, pruning attention heads and intermediate dimensions to form slimmer subnetworks, and then *depth-adaptive training*, progressively pruning layers with knowledge distillation. Denote by $f_{\theta(w,d)}$ a subnetwork with width $w$ and depth $d$. DynaBERT jointly optimizes

$$\min_{\{\theta^{(w,d)}\}} \ \mathbb{E}_{(w,d) \sim \mathcal{U}}\, \mathcal{L}(f_{\theta(w,d)}(x), y) + \mu\, \mathcal{L}_{\text{KD}}(f_{\theta(w,d)}(x), f_{\theta^{\text{full}}}(x))\,,$$

where $\mathcal{L}_{\text{KD}}$ denotes a knowledge-distillation loss from the full teacher model.

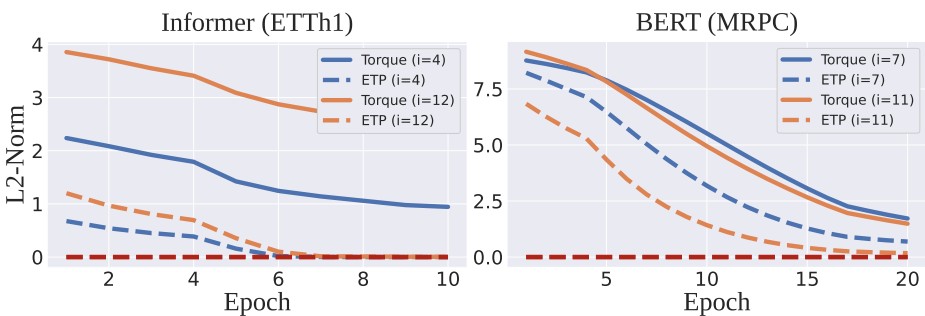

Figure 5: Additional results on the l2-norm analysis during the training process.

3. **EBERT:** It introduces an input-adaptive structured pruning approach for transformers, in which lightweight predictors produce binary masks over attention heads and FFN channels conditioned on the [CLS] token representation. The masks are sampled with a Gumbel–Softmax relaxation for differentiable optimization. The objective balances task accuracy with FLOPs constraints:

$$\mathcal{L} = \mathcal{L}_{\text{task}} + \lambda_1 \left( \frac{F_c}{F_o} - C_t \right)^2 + \lambda_2 \left( L_M + L_F \right),$$

where $F_o$ and $F_c$ denote original and current FLOPs, $C_t$ is a target budget, and $L_M, L_F$ penalize unbalanced pruning across attention and feed-forward modules.

4. **LLM-Pruner:** The method first constructs *dependency groups*, i.e., coupled parameter sets that must be pruned jointly to maintain architectural validity. For each group, an importance score is estimated using a first-order Taylor expansion with Hessian-based correction under a limited data budget. Formally, letting $\mathcal{G}$ denote the set of groups and $I(g)$ their importance,

$$\min_{\mathcal{M} \subseteq \mathcal{G}} \sum_{g \in \mathcal{M}} I(g) \quad \text{s.t. dependency closure,}$$

where $\mathcal{M}$ is the set of groups selected for removal. Following pruning, lightweight post-training (e.g. LoRA) efficiently recovers accuracy.

## 7.3 ADDITIONAL EXPERIMENTS ON L2-NORM LEARNING PROCESS

We further present additional results of the L2-norm analysis during training process on Informer (ETTh1) and BERT (MRPC) as a supplementary of RQ1. The detailed results are shown in Figure 5. It is obvious that the results are consistent with that of RQ1. For example, for Informer trained and evaluated on ETTh1, ETP manages to optimally prune both investigated modules $(m_4^l, m_{12}^l)$ as ETP deems them redundant for effective inference (i.e., L2-norm equals to 0 ($||m_4^l|| = 0.0, \; ||m_{12}^l|| = 0.0$)), while the ones regularized by the vanilla Torque remains a high L2-norm for these modules (i.e., $||m_4^l|| = 0.99, \; ||m_{12}^l|| = 2.44$). The extensive L2-norm analysis during the training process validate that the exponential force application scheme can indeed help ETP achieve a much sparser neural network architecture, and therefore achieve a much higher compression rate with lower performance drop.

## 7.4 DIFFERENT FORCE SCHEMES

A key motivation of Exponential Torque Pruning (ETP) is to address the sub-optimal regularization exhibited by linear or constant force-application schemes. As shown in Figure 2, even after applying linear regularization (as in Torque), the resulting sparsity pattern remains sub-optimal.

To overcome this limitation, we propose using the Heaviside step function as the force-application scheme:

- Modules close to the pivot (i.e., essential for inference) receive zero penalty.
- Distant, redundant modules incur a large penalty.

Table 8: Comparison of different force schemes on VGG19 (CIFAR-100, base accuracy 73.50%).

| Speed-Up | ETP (Ours) | Torque | Log-Torque | Group Lasso (L2) |
|---|---|---|---|---|
| 3× | **73.93 ± 0.17** | 72.40 ± 0.29 | 71.39 ± 0.20 | 71.49 ± 0.14 |
| 6× | **72.13 ± 0.13** | 70.86 ± 0.03 | 70.25 ± 0.07 | 70.36 ± 0.12 |
| 9× | **71.54 ± 0.19** | 65.87 ± 0.12 | 66.13 ± 0.11 | 66.01 ± 0.04 |
| 15× | **69.95 ± 0.08** | 62.31 ± 0.17 | 61.84 ± 0.29 | 61.74 ± 0.03 |
| 24× | **67.98 ± 0.11** | 60.08 ± 0.04 | 58.11 ± 0.78 | 58.29 ± 0.11 |

Since the Heaviside function is non-differentiable, we approximate it with a differentiable exponential function, enabling gradient-based optimization.

We compare ETP against the following alternative force application schemes, which include:

1. **Torque:** Serves as one of our general-purpose benchmarks, with the loss defined as

$$\mathcal{L}_{\text{tot}} = \mathcal{L}_{\text{task}} + \lambda_T \sum_n \|T_n\|_2,$$

   where

$$\|T_n\|_2 = \|W_n\|_2 \cdot |r_n - r_p|.$$

   Here, the regularization grows linearly with the distance from the pivot, and its effect can be directly modulated by $\lambda_T$.

2. **Log-Torque:** Defined as

$$\mathcal{L}_{\text{tot}} = \mathcal{L}_{\text{task}} + \lambda_T \sum_n \|T_n\|_2,$$

   where

$$\|T_n\|_2 = \|W_n\|_2 \cdot \log(|r_n - r_p|).$$

   In this variant, the force grows logarithmically with distance, resulting in a slower increase compared to the linear Torque scheme.

3. **Group-Lasso:** Defined as

$$\mathcal{L}_{\text{tot}} = \mathcal{L}_{\text{task}} + \lambda_T \sum_n \|T_n\|_2,$$

   where

$$\|T_n\|_2 = \|W_n\|_2 \cdot 1^{|r_n - r_p|}.$$

   In this case, the penalty is independent of the distance from the pivot, effectively reducing to standard $\ell_2$ regularization on the filter weights.

As shown in Table 8, ETP consistently outperforms all competing schemes on VGG19 (CIFAR-100) across different speed-up ratios, achieving higher accuracy at every compression level. Notably, the Log-Torque variant underperforms the baseline Torque scheme, highlighting the need for stronger force scheme.

## 7.5 STATISTICAL VALIDATION

To ensure the statistical significance of our findings, we conduct 5 independent runs and apply paired $t$-tests against the strongest competing baseline in each setting. We select a few tasks to run statistical significance tests, namely:

- BERT @ SST-2,
- RoBERTa @ MRPC,
- GAT @ PPI (9× and 12×),
- VGG19 @ CIFAR-100 (9×).

Table 9: Statistical validation on BERT (SST-2) and RoBERTa (MRPC).

| Dataset | Model | Method | Accuracy $\pm$ Std | Speed-Up | $p$-value vs ETP |
|---------|-------|--------|--------------------|----------|------------------|
| SST-2 | BERT | ETP | $92.13 \pm 0.01$ | $11\times$ | - |
| | | DepGraph | $91.86 \pm 0.05$ | $11\times$ | 0.009 |
| | | Torque | $90.81 \pm 0.06$ | $11\times$ | 0.005 |
| MRPC | RoBERTa | ETP | $86.57 \pm 0.01$ | $8\times$ | - |
| | | DepGraph | $86.01 \pm 0.06$ | $8\times$ | 0.003 |
| | | Torque | $85.27 \pm 0.12$ | $8\times$ | 0.003 |

Table 10: Statistical validation on VGG19 (CIFAR-100) at $9\times$ speed-up.

| Method | Accuracy $\pm$ Std | Speed-Up | $p$-value vs ETP |
|--------|--------------------|----------|------------------|
| ETP (Ours) | $71.54 \pm 0.19$ | $9\times$ | - |
| DepGraph | $70.38 \pm 0.32$ | $9\times$ | 0.010 |
| Torque | $65.77 \pm 0.30$ | $9\times$ | 0.000 |
| GReg-1 | $67.35 \pm 0.22$ | $9\times$ | 0.000 |
| GReg-2 | $67.55 \pm 0.29$ | $9\times$ | 0.000 |

Table 11: Statistical validation on GAT (PPI dataset) at $9\times$ and $12\times$ speed-ups.

| Method | F1 $\pm$ Std | Speed-Up | $p$-value vs ETP |
|--------|--------------|----------|------------------|
| ETP | $0.9633 \pm 0.0005$ | $9\times$ | - |
| DepGraph | $0.9610 \pm 0.0000$ | $9\times$ | 0.015 |
| ETP | $0.9587 \pm 0.0005$ | $12\times$ | - |
| DepGraph | $0.9555 \pm 0.0007$ | $12\times$ | 0.004 |

As shown in Tables 9–11, ETP yields statistically significant improvements in all cases ($p < 0.05$). For example, on SST-2 with BERT, ETP improves accuracy by 0.27 points over DepGraph ($p = 0.009$). On VGG19 ($9\times$ speed-up), ETP surpasses the closest baseline by more than 1 percentage point ($p = 0.010$). These results confirm that ETP's gains are consistent and not due to variance.

## 7.6 Ablation of $\lambda$ and $\beta$

We conduct ablation experiments to study the sensitivity of our method to the hyperparameters $\lambda$ and $\beta$. Recall that $\lambda$ is defined as $\lambda = \exp(a/|G_x|)$, where $a$ controls the steepness of the exponential weighting, while $\beta$ governs the relative importance of the ETP loss term.

**Effect of $\lambda$:** Table 12 shows results on **VGG19 (CIFAR100)** and **ResNet50 (ImageNet-1k)**. Varying $a$ from 2.5 to 15 has little impact on the speed-up ratio, which remains constant ($9\times$ for VGG19 and $2.3\times$ for ResNet50). However, accuracy steadily improves as $a$ increases. For instance, on CIFAR100, accuracy rises from 68.19% to 72.47% as $a$ increases from 2.5 to 15. Similarly, on ImageNet-1k, accuracy improves from 73.55% to 76.04%. This trend suggests that a larger $a$ leads to a sharper exponential curve, causing the force application to approximate a Heaviside function more closely, thereby offering modest but consistent performance gains.

Table 12: Ablation of $\lambda$: varying $a$ on VGG19 (CIFAR100) and ResNet50 (ImageNet-1k).

| VGG19 on CIFAR100 | $a = 2.5$ | $a = 5$ | $a = 7.5$ | $a = 10$ | $a = 15$ |
|---|---|---|---|---|---|
| Speed-Up | $9\times$ | $9\times$ | $9\times$ | $9\times$ | $9\times$ |
| Acc. (%) | 68.19 | 71.30 | 71.50 | 71.90 | 72.47 |

| ResNet50 on ImageNet-1k | $a = 2.5$ | $a = 5$ | $a = 7.5$ | $a = 10$ | $a = 15$ |
|---|---|---|---|---|---|
| Speed-Up | $2.3\times$ | $2.3\times$ | $2.3\times$ | $2.3\times$ | $2.3\times$ |
| Acc. (%) | 73.55 | 75.17 | 75.62 | 75.63 | 76.04 |

**Effect of $\beta$:** Table 13 presents the results for **VGG19 on CIFAR100**. Increasing $\beta$ directly impacts the speed-up ratio, which scales from $3.69\times$ at $\beta = 10^{-5}$ to $24\times$ at $\beta = 3 \times 10^{-3}$. This confirms that $\beta$ strongly controls the aggressiveness of the pruning process. However, this comes with a trade-off in accuracy: performance peaks at 73.93% for $\beta = 10^{-5}$ and declines to 67.98% at the highest value. Thus, while larger $\beta$ enables more aggressive acceleration, it must be chosen carefully to balance accuracy and efficiency.

Table 13: Ablation of $\beta$: varying $\beta$ on VGG19 (CIFAR100).

| $\beta$ | $1 \times 10^{-5}$ | $1 \times 10^{-4}$ | $5 \times 10^{-4}$ | $1 \times 10^{-3}$ | $3 \times 10^{-3}$ |
|---|---|---|---|---|---|
| Speed-Up | $3.69\times$ | $6\times$ | $9\times$ | $14\times$ | $24\times$ |
| Acc. (%) | 73.93 | 72.53 | 71.30 | 69.95 | 67.98 |

## 7.7 HARDWARE EFFICIENCY ANALYSIS

While MACs-based reductions are a standard proxy for computational savings in structured pruning, they do not always translate proportionally to real deployment gains due to hardware-, kernel-, and memory-related overheads. To provide a more faithful assessment, we additionally measure the **wall-clock inference time speed-up**, defined as the ratio between the dense and pruned model runtimes on the entire test set.

As shown in Table 14, ETP yields substantial end-to-end inference acceleration across convolutional, transformer, and graph neural network architectures. Importantly, the measured speed-ups closely track the theoretical MACs-based estimates, confirming that the spatial sparsity patterns induced by ETP are highly aligned with the underlying hardware execution pathways.

Table 14: Wall-clock inference speed-up versus MACs-based theoretical speed-up. Pruned accuracy is reported with the dense baseline accuracy in parentheses.

| Model & Dataset | Pruned Acc. | Inference Speed-Up | MACs Speed-Up |
|---|---|---|---|
| VGG-19 on CIFAR100 (73.5%) | 71.3 | $6.82\times$ | $9\times$ |
| BERT on SST-2 (93.5%) | 92.1 | $7.41\times$ | $11\times$ |
| RoBERTa on MRPC (90.0%) | 86.6 | $5.72\times$ | $8\times$ |
| GAT on PPI (0.986) | 0.963 | $5.48\times$ | $9\times$ |

Across all settings, ETP consistently provides strong inference-time gains while preserving model quality. This confirms the practicality of ETP for both real-time deployment and large-scale inference workloads. To address increasing interest in deployment gains, we further benchmark wall-clock latency and energy consumption across *five* hardware platforms: NVIDIA A100, L4, RTX 8000, Tesla T4, and Google TPU v6. Table 15 reports results for two representative workloads (BERT on SST-2 and VGG-19 on CIFAR-100).

ETP achieves:

- **6.4–9.3$\times$ latency speed-up** on GPUs and TPUs,

Table 15: Cross-hardware evaluation of latency and energy for pruned BERT (SST-2) and VGG-19 (CIFAR-100). Theoretical speed-ups are $11\times$ (BERT) and $9\times$ (VGG-19).

| Model | Hardware | Base Lat. (ms) | Pruned Lat. (ms) | Speedup | Base Energy (J) | Pruned Energy (J) | Reduction |
|---|---|---|---|---|---|---|---|
| BERT SST-2 (92.1%) | NVIDIA A100 | 45.863±0.115 | 7.098±0.377 | 6.46× | 14.471±0.605 | 2.439±0.722 | 83.1% |
| | NVIDIA L4 | 101.166±1.258 | 10.868±0.341 | 9.31× | 7.492±0.082 | 0.787±0.019 | 89.5% |
| | Quadro RTX 8000 | 66.540±0.608 | 9.650±0.044 | 6.90× | 17.547±0.173 | 3.507±0.551 | 80.0% |
| | NVIDIA Tesla T4 | 188.705±3.687 | 23.293±0.486 | 8.10× | 14.585±0.586 | 1.622±0.079 | 88.9% |
| | Google TPU v6 | 546.013±12.463 | 79.637±8.126 | 6.86× | 73.390±4.653 | 11.277±3.972 | 84.6% |
| VGG-19 CIFAR-100 (71.30%) | NVIDIA A100 | 5.578±0.004 | 0.818±0.013 | 6.82× | 1.853±0.559 | 0.278±0.119 | 85.3% |
| | NVIDIA L4 | 14.914±0.151 | 6.117±0.084 | 4.43× | 1.080±0.010 | 0.240±0.000 | 77.4% |
| | Quadro RTX 8000 | 15.013±0.065 | 3.538±0.804 | 4.25× | 3.535±0.023 | 1.556±0.222 | 56.0% |
| | NVIDIA Tesla T4 | 35.426±0.192 | 9.262±0.116 | 3.82× | 2.521±0.273 | 1.090±0.230 | 56.7% |
| | Google TPU v6 | 171.202±6.831 | 19.641±0.373 | 8.72× | 22.934±2.0142 | 2.630±0.371 | 88.5% |

- **56–90% energy reduction** depending on device class,
- close alignment to theoretical sparsity-induced MACs reductions.

These results highlight that ETP maintains efficiency across both high-end accelerators (A100, TPU v6) and cost-efficient inference hardware (L4, T4), demonstrating broad practicality for deployment scenarios.

## 7.8 ADDITIONAL EXPERIMENTS ON LLMs

To further validate the generality of ETP beyond vision models, we evaluate its performance on large language models of both decoder-only and encoder-based architectures. Decoder models are assessed using perplexity—a direct measure of generative modeling capability—while encoder models are evaluated on downstream tasks from the GLUE benchmark. Together, these experiments examine whether the sparsity patterns induced by ETP preserve both intrinsic language modeling behavior and task-specific semantic reasoning.

**Decoder-Based LLMs (Llama-3-8B).** Table 16 reports pruning results on Llama-3-8B under three sparsity regimes: 50% unstructured sparsity, 4:8 semi-structured sparsity, and the more restrictive 2:4 pattern. Perplexity (PPL) is used as the evaluation metric; lower values indicate better preservation of the next-token distribution.

Across all regimes, ETP consistently outperforms existing unstructured and semi-structured pruning baselines. At 50% sparsity, ETP achieves a perplexity of 5.84—nearly identical to the dense model's 5.72—while substantially surpassing classical baselines such as Magnitude (15.21) and SparseGPT (7.06). More recent structured and correlation-aware approaches, including SlimGPT (11.41), FLAP (9.30), and PP (6.81), also perform worse than ETP, indicating that exponential regularization provides a more effective inductive bias for identifying redundant components.

In the 4:8 semi-structured case, ETP again delivers the strongest performance with a perplexity of 6.27, compared to 8.46 (Wanda) and 8.01 (SparseGPT). Even under the highly restrictive 2:4 constraint—where pruning decisions are tightly coupled to hardware-imposed sparsity patterns—ETP achieves 9.71, a notable improvement over Wanda (11.02) and SparseGPT (10.53). These results highlight that ETP preserves attention heads and MLP channels critical to autoregressive modeling, while reliably eliminating peripheral structures that contribute little to next-token prediction. The robustness of ETP across

Table 16: Pruning results on Llama-3-8B. Lower perplexity is better.

| Method | Sparsity | Perplexity |
|---|---|---|
| **Llama-3-8B** | | |
| Dense | - | 5.72 |
| Magnitude | 50% | 15.21 |
| Wanda | 50% | 6.97 |
| SparseGPT | 50% | 7.06 |
| SlimGPT | 50% | 11.41 |
| FLAP | 50% | 9.30 |
| PP | 50% | 6.81 |
| **ETP (Ours)** | **50%** | **5.84** |
| Magnitude | 4:8 | 16.98 |
| Wanda | 4:8 | 8.46 |
| SparseGPT | 4:8 | 8.01 |
| **ETP (Ours)** | **4:8** | **6.27** |
| Magnitude | 2:4 | 55.37 |
| Wanda | 2:4 | 11.02 |
| SparseGPT | 2:4 | 10.53 |
| **ETP (Ours)** | **2:4** | **9.71** |

Table 17: Performance comparison between BERT-base and an ETP-pruned BERT on GLUE tasks. Despite aggressive 86.89% parameter reduction, the pruned model maintains strong downstream performance.

| Dataset | Metric | BERT-base | Pruned BERT | Drop | Speedup |
|---------|--------|-----------|-------------|------|---------|
| SST-2 | Accuracy | 0.935 | 0.9256 | $-0.0094$ | $9\times$ |
| MRPC | F1 | 0.880 | 0.8440 | $-0.0360$ | $9\times$ |
| STS-B | Pearson | 0.889 | 0.8620 | $-0.0270$ | $9\times$ |
| QQP | F1 | 0.887 | 0.8521 | $-0.0349$ | $9\times$ |
| MNLI | Accuracy | 0.843 | 0.8031 | $-0.0399$ | $9\times$ |
| QNLI | Accuracy | 0.905 | 0.8554 | $-0.0496$ | $9\times$ |
| RTE | Accuracy | 0.711 | 0.6859 | $-0.0251$ | $9\times$ |
| WNLI | Accuracy | 0.653 | 0.5634 | $-0.0896$ | $9\times$ |

sparsity formats underscores its advantage as a principled, architecture-aware pruning strategy for large decoder-only transformers.

**Encoder-Based LLMs (BERT-base on GLUE).** To assess the downstream reasoning capability of ETP-pruned models, we apply ETP to BERT-base and evaluate on multiple GLUE tasks (Table 17). Despite an 86.89% parameter reduction and a $9\times$ inference speed-up, the pruned model preserves competitive task performance across classification, entailment, and semantic similarity benchmarks.

Performance on sentiment and paraphrase detection tasks (SST-2, MRPC, QQP) remains stable, demonstrating that lexical and syntactic reasoning pathways remain intact. Tasks requiring fine-grained semantic inference (MNLI, QNLI, RTE) exhibit moderate drops yet remain within a practical operating range given the extreme compression level. These results suggest that ETP retains the most influential attention heads and MLP channels driving contextual representation quality, while safely pruning peripheral structures that contribute less to downstream task performance.

## 7.9 ETP AND QUANTIZATION

Modern deployment scenarios increasingly require models that are both sparse and low-precision in order to meet stringent latency, memory, and energy constraints. To assess whether ETP is compatible with quantization pipelines, we evaluate dense, ETP-pruned, and ETP+QAT variants across CNN and Transformer architectures. In addition to accuracy and inference speed, Table 18 reports the corresponding on-disk model sizes, allowing us to quantify the combined compression effect of pruning and quantization.

Table 18: Accuracy, speed-up, and model size of ETP-pruned models with and without QAT. Model size refers to the on-disk checkpoint size in megabytes (MB).

| Model | Dataset | Accuracy (%) | | | Speed-up | Model Size (MB) | | |
|-------|---------|------|-----|---------|----------|------|-----|---------|
| | | Base | ETP | ETP+QAT | | Base | ETP | ETP+QAT |
| ResNet-56 | CIFAR-10 | 93.44 | 93.56 | 93.38 | $2.72\times$ | 3.42 | 1.17 | 0.29 |
| VGG-19 | CIFAR-100 | 73.50 | 71.30 | 71.03 | $9\times$ | 80.16 | 7.13 | 1.78 |
| ViT-B/16 | ImageNet-1K | 81.07 | 81.93 | 80.52 | $1.69\times$ | 346.27 | 174.88 | 43.27 |

Across all three architectures, ETP alone yields substantial storage reductions while preserving competitive accuracy. For instance, ResNet-56 is reduced from 3.42 MB to 1.17 MB (a 65.8% reduction), while achieving a slight improvement in accuracy. VGG-19 undergoes an order-of-magnitude shrinkage (80.16 MB $\rightarrow$ 7.13 MB), reflecting the significant intra-filter redundancy that ETP removes. ViT-B/16, despite operating at a much larger scale, also compresses by nearly half (346.27 MB $\rightarrow$ 174.88 MB) while exhibiting improved top-1 accuracy, suggesting that ETP acts as an effective regularizer even on transformer architectures.

When quantization-aware training is additionally applied, the memory footprint decreases even more dramatically. For example, ResNet-56 shrinks from 3.42 MB to only 0.29 MB, and VGG-19 from 80.16 MB to 1.78 MB—representing over a **45×** compression relative to the dense baseline. ViT-B/16 also benefits substantially, with the ETP+QAT variant occupying only 43.27 MB, an 87.5% reduction from the dense model. Crucially, these large reductions come with only modest accuracy changes: 93.56% → 93.38% for ResNet-56 and 81.93% → 80.52% for ViT-B/16.

These results highlight that ETP not only produces structured sparsity patterns that preserve model accuracy and reduce compute, but also yields weight distributions that are inherently stable under quantization. By removing redundant or low-signal parameter groups prior to quantization, ETP minimizes the quantization error accumulated in critical regions of the network, thereby enabling aggressive bitrate reduction with minimal degradation. The synergy between ETP and QAT holds across both convolutional and transformer-based models, demonstrating that ETP serves as a strong foundation for multi-dimensional compression pipelines targeting latency, compute, and memory simultaneously.

### 7.10 SUBGRADIENT STATIONARITY ANALYSIS OF ETP

To formalize the effect of exponential distance weighting, we analyze the first-order stationarity (KKT) conditions of the regularized objective

$$J(w) = L(w) + \beta \sum_g R(w_g),$$

where $L(w)$ is the task loss, $w_g$ denotes the parameters of group $g$, and $R(w_g)$ is a distance-weighted group regularizer. We compare three choices:

$$\text{Group Lasso: } R(w_g) = \|w_g\|_2, \qquad \text{Torque (Linear): } R(w_g) = d(g)\|w_g\|_2,$$

$$\text{ETP (Exponential): } R(w_g) = \lambda^{d(g)}\|w_g\|_2, \ \lambda > 1,$$

where $d(g)$ is the distance of group $g$ from a designated pivot (e.g., kernel center or early-layer position).

Because the group norm is non-differentiable at $w_g = 0$, we use subgradient analysis. At any stationary point $w^\star$, the optimality condition requires

$$0 \in \nabla_{w_g} L(w^\star) + \beta\, c_g\, \partial\|w_g^\star\|_2,$$

where $c_g$ is the distance-dependent weight. For groups that are driven exactly to zero ($w_g^\star = 0$), this condition reduces to the well-known pruning criterion:

$$\|\nabla_{w_g} L(w^\star)\|_2 \leq \beta\, c_g.$$

Thus, the effective pruning threshold for group $g$ is

$$T(g) = \beta\, c_g.$$

**Comparison of Regularizers.** Group Lasso uses a constant threshold $T(g) = \beta$, which forces pruning uniformly across all distances; increasing $\beta$ risks removing high-importance (small-$d$) groups and leads to underfitting. Torque introduces linear scaling $T(g) = \beta d(g)$, but the pruning-force ratio between far and near groups grows only as $d(\text{far})/d(\text{near})$, which is often insufficient in deep networks.

ETP instead uses an exponentially increasing threshold $T(g) = \beta\lambda^{d(g)}$. The pruning-force ratio between far and near groups becomes

$$\frac{T(\text{far})}{T(\text{near})} = \lambda^{d(\text{far})-d(\text{near})},$$

which grows exponentially in the distance difference. Consequently, near groups ($d(g)$ small) have low thresholds and are rarely pruned, while far groups rapidly exceed the pruning condition and are removed. This produces a *step-like* separation—a smooth convex surrogate for a hard distance-based prior—and naturally induces strong sparsity heterogeneity across spatial or architectural depth.

Overall, this analysis explains why ETP consistently preserves core functional components while aggressively suppressing distant or redundant groups, aligning well with the empirical sparsity patterns observed across CNNs, ViTs, and LLMs.

