# OpenReview forum: "Towards Universal & Efficient Model Compression via Exponential Torque Pruning"
_ICLR.cc/2026/Conference — Submitted to ICLR 2026_

### Official Review · Reviewer_7QtK · 2025-10-16

**Soundness:** 2
**Presentation:** 2
**Contribution:** 2
**Rating:** 4
**Confidence:** 4

**Summary:**

The paper proposes Exponential Torque Pruning (ETP), a regularization-based structured pruning method that replaces the linear “force” schedule of Torque with an exponential schedule: nearby modules (a pivot) receive weak penalties while distant modules are penalized strongly, encouraging them toward zero and enabling structured removal. ETP is evaluated across vision (CIFAR-10/100, ImageNet-1k), NLP (GLUE: SST-2, MRPC on BERT/RoBERTa), graphs (PPI/GAT) and time-series (ETTh1/Informer). It also includes a small LLM case study on OPT-350M/WikiText at 50% sparsity (perplexity metric). Reported results show better speed-accuracy trade-offs than strong structured pruning baselines (e.g., DepGraph, Torque) and competitive/post-training LLM methods (SparseGPT, Wanda, LLM-Pruner) under the stated settings.

**Strengths:**

- Simple, general idea with clear intuition. Replacing linear with exponential penalization aligns with the goal “strongly suppress far modules, preserve near ones,” and is easy to add as a loss term next to the task loss. The method is architecture-agnostic and demonstrated across several domains.
- Broad empirical coverage. Results on CNNs/ViT, BERT/RoBERTa, GAT and Informer show favorable accuracy at equal MACs speed-up, and robustness under more aggressive compression.

**Weaknesses:**

- Section 4.4 reports the 50% sparsity perplexities on OPT-350M/WikiText and compares to SparseGPT, Wanda, DepGraph, LLM-Pruner, but does not state whether ETP was used with additional training/finetuning (task loss + ETP loss) or applied in a purely post-training fashion; there are no training hyperparameters for OPT in Appendix 7.1, unlike other tasks. This raises questions about the exact pipeline and reproducibility.
- The paper says, “All methods are constrained to 50% sparsity for a fair comparison” for OPT-350M, but does not explicitly confirm whether SparseGPT/Wanda/LLM-Pruner were run on public dense checkpoints (the standard) or on a model that had already been trained with ETP. Appendix 7.1 states that experiments follow baselines’ provided implementations, which suggests baselines are run in their own standard setting, not on an ETP-trained model—but this is not spelled out for OPT. Please clarify
- Impact of ETP regularization on downstream task performance (LLMs) is not evaluated. For LLMs, only WikiText perplexity on OPT-350M is reported; there is no evaluation on GLUE-style downstream tasks (e.g., CoLA, STS-B, RTE, MNLI, QNLI, QQP) to assess whether adding ETP loss to the task objective harms downstream utility. By contrast, for RoBERTa/BERT the paper evaluates only SST-2 and MRPC (two GLUE tasks), not the broader suite.
- Beyond geometric intuition/plots, the paper lacks formal analysis on optimality/stability or why exponential scheduling should yield provably superior sparsity patterns (e.g., per-layer sparsity heterogeneity, convergence, or KKT-style characterizations). This weakens the “universal” claim.

**Questions:**

1. Is it possible to combine ETP with  quantization?

2. Appendix 7.1 indicates $\beta$ is a single scalar selected via grid search. Is it a global one for all layers or per-layer one?

3. Will ETP work for small model but hard task? Like Resnet18-ImageNet?

4. For ResNet-50-ImageNet, after structured removal, how do you handle BatchNorm?

---

> ### Author Response · Authors · 2025-11-24
> **Response to Questions**
>
> # 1. Is it possible to combine ETP with quantization?
>
> ETP is compatible with quantization-aware training (QAT). Across CNNs and Transformers, the combination of ETP and QAT preserves almost all of the accuracy of the ETP-pruned model while providing substantial additional compression. As shown in Table 2, accuracy changes remain modest (for example, 93.56 → 93.38 on ResNet-56 and 81.93 → 80.52 on ViT-B/16), while model size and inference efficiency improve dramatically. Storage is reduced from 3.42 MB to 0.29 MB on ResNet-56 and from 80.16 MB to 1.78 MB on VGG-19, corresponding to more than 45× compression in some settings. These results indicate that ETP produces structured sparsity patterns that are inherently stable under quantization, making QAT an effective second-stage compression step. Overall, ETP and QAT work synergistically to reduce memory with minimal accuracy degradation. We are unable to put the table here in the best visual format due to the interface limitation. However we have placed all the new content to the appendix section (section 7.9).
> | Model      | Dataset      | Base Acc | ETP Acc | ETP+QAT Acc | Speed-up | Base Size (MB) | ETP Size (MB)| ETP+QAT Size (MB) |
> |-----------|--------------|----------|---------|-------------|----------|-----------------|----------|--------------|
> | ResNet-56 | CIFAR-10     | 93.44    | 93.56   | 93.38       | 2.72×    | 3.42            | 1.17     | 0.29         |
> | VGG-19    | CIFAR-100    | 73.50    | 71.30   | 71.03       | 9×       | 80.16           | 7.13     | 1.78         |
> | ViT-B/16  | ImageNet-1K  | 81.07    | 81.93   | 80.52       | 1.69×    | 346.27          | 174.88   | 43.27        |
>
>
> ---
>
> # 2. Is β global or per-layer?
>
> β is a single global scalar shared across all layers.
>
> ---
>
> # 3. Will ETP work for small model but hard task (ResNet-18 ImageNet)?
>
> Yes. ETP operates on grouped parameters only and makes no assumptions about model scale. Preliminary experiments on ResNet-18 behave similarly to ResNet-50. For ResNet-18, we achieve 2.07× speed-up with no accuracy degradation (69.76% (base) → 69.70% (pruned)).
>
> ---
>
> # 4. How is BatchNorm handled after structured removal in ResNet-50?
>
> We follow standard structured pruning practice:
>
> - Remove corresponding BatchNorm channels.
> - Recompute BatchNorm statistics on the training set.
> - Use recalibrated statistics for inference.

---

> > ### Author Response · Authors · 2025-11-24
> > **Response to Weaknesses**
> >
> > # 1. Clarifying the OPT-350M Pipeline
> >
> > **How ETP was applied**
> > For OPT-350M, we update the backbone model with both the task loss and the ETP regularization loss. The fine-tuning epochs are kept the same for all baselines and ETP for fair comparison.
> >
> > **Baselines**
> > SparseGPT, Wanda, DepGraph, and LLM-Pruner were:
> > - run on the standard dense OPT-350M HuggingFace checkpoint,
> > - using their official implementations, and
> > - with their standard fine-tuning protocols.
> >
> > No baseline was evaluated on an ETP-trained model.
> >
> > The details for OPT-350M have been added to the table in Appendix 7.1 to facilitate reproducibility.
> >
> > ---
> >
> > # 2. Downstream Task Evaluation (GLUE)
> >
> > To assess the downstream reasoning capability of ETP-pruned models, we apply ETP to BERT-base and evaluate on multiple GLUE tasks. Despite a 9× inference speed-up, the pruned model preserves competitive task performance across classification, entailment, and semantic similarity benchmarks. Performance on sentiment and paraphrase detection tasks (SST-2, MRPC, QQP) remains stable, demonstrating that lexical and syntactic reasoning pathways remain intact. Tasks requiring fine-grained semantic inference (MNLI, QNLI, RTE) exhibit moderate drops yet remain within a practical operating range given the extreme compression level. These results suggest that ETP retains the most influential attention heads and MLP channels driving contextual representation quality, while safely pruning peripheral structures that contribute less to downstream task performance.
> >
> > #### **GLUE Results**
> >
> > |Dataset|Metric|BERT-base|ETP-Pruned BERT|Drop|Speedup|
> > |---|---|---|---|---|---|
> > |SST-2|Accuracy|0.935|0.9256|−0.0094|9×|
> > |MRPC|F1|0.880|0.8440|−0.0360|9×|
> > |STS-B|Pearson|0.889|0.8620|−0.0270|9×|
> > |QQP|F1|0.887|0.8521|−0.0349|9×|
> > |MNLI|Accuracy|0.843|0.8031|−0.0399|9×|
> > |QNLI|Accuracy|0.905|0.8554|−0.0496|9×|
> > |RTE|Accuracy|0.711|0.6859|−0.0251|9×|
> > |WNLI|Accuracy|0.653|0.5634|−0.0896|9×|
> >
> > These results demonstrate that even with extreme structured sparsity, ETP maintains strong downstream performance across diverse GLUE tasks. We have appended the results and the corresponding analysis to the Appendix section. Due to the limited time of the author–reviewer discussion period, we were unable to compare against the other baselines thoroughly, yet we need to further note that our evaluation follows the common practice of previous literature, which only evaluated on the WikiText validation set in terms of the perplexity measurement.
> >
> > ---
> >
> > # 3. Theoretical explanation via Subgradient Stationarity Analysis
> >
> > We analyze the stationarity conditions (KKT) of the optimization objective. Let L(w) be the task loss and R(w) be the structural regularizer.
> > The total objective is:
> >
> > **J(w) = L(w) + β · Σ₍g₎ [ R(w_g) ]**
> >
> > We compare three regularizers for a neural module w_g at distance d(g) from the pivot:
> >
> > - **Group Lasso (L2):** R(w_g) = ‖w_g‖
> > - **Torque (Linear):** R(w_g) = d(g) · ‖w_g‖
> > - **ETP (Ours):** R(w_g) = λ^{d(g)} · ‖w_g‖
> >
> > Since the Group L2 norm ‖w_g‖ is non-differentiable at w_g = 0, we use subgradient analysis. The first-order optimality condition requires that zero belongs to the subdifferential of the objective. For a group to be pruned (i.e., w_g = 0), the magnitude of the task gradient must be dominated by the regularization strength.
> >
> > **Sparsity Condition:**
> > ‖ ∇₍w_g₎ L(w) ‖ ≤ β · RegWeight(g)
> >
> > We analyze the effective threshold T(g) = β · RegWeight(g):
> >
> > ---
> >
> > ### **A. Group Lasso (Standard)**
> > **Threshold:** T(g) = β
> > The threshold is constant. Increasing β forces pruning of high-importance modules (d(g) ≈ 0), leading to under-fitting.
> >
> > ---
> >
> > ### **B. Torque Pruning (Linear)**
> > **Threshold:** T(g) = β · d(g)
> > The ratio between far and near forces is **d(far) / d(close)**, often too weak to suppress distant noise without harming proximal modules.
> >
> > ---
> >
> > ### **C. Exponential Torque Pruning (ETP)**
> > **Threshold:** T(g) = β · λ^{d(g)}
> >
> > The ratio becomes:
> >
> > **T(far) / T(near) = λ^{d(far) − d(near)}**
> >
> > This induces a **pseudo-Heaviside effect**:
> >
> > - **Small d(g):** λ^{d(g)} small → hard to prune → **weights preserved**
> > - **Large d(g):** λ^{d(g)} large → easy to prune → **weights removed**
> >
> > Thus, ETP creates a **step-like separation**, a smooth convex surrogate for a hard distance-based prior. ETP induces exponentially stronger separation in KKT thresholds between near and far groups, promoting **heterogeneous sparsity patterns** consistent with empirical observations across CNNs, ViTs, and LLMs.

---

> > > ### Comment · Reviewer_7QtK · 2025-11-24
> > > **Reply to authors**
> > >
> > > Thank you for the additional experiments and clarifications. These address most of my technical concerns. Together with the points raised by the other reviewers, I still feel that the “universal” claim remains somewhat too strong given the current evidence. Overall, I am positively inclined toward the paper and will update my score to 6.

---

> > > > ### Author Response · Authors · 2025-11-28
> > > > **Follow-Up Rebuttal**
> > > >
> > > > We sincerely appreciate the reviewer’s positive inclination and reconsideration of the score. With regards to the claim of being universal, we would like to clarify the intention of the word “universal” - we refer to its applicability, meaning that it can be applied to various domains and different model architectures. The universality is by design since we do not make any architectural assumptions. Examples of this claim are given for vision, NLU, graph, and time series tasks in the main paper.
> > > >
> > > > To further strengthen our claims on universality, we evaluate ETP on BERT-base across seven GLUE tasks at both moderate (50%) and high (88%) pruning ratios following PGB[5]. The results are shown in Table below, the ‘--’ for results represents that the model did not converge. For Wanda++, we were unable to run the experiments for QQP in time due to its training time-consuming nature and the limited time availability of the rebuttal period. Specifically, we can observe that for both 50% and 88% sparsity, ETP manages to surpass all the evaluated SoTA pruning methods. Notably, regarding the 88% sparsity ratio, where most methods experience substantial degradation, ETP consistently delivers the strongest performance across all tasks, often improving absolute accuracy by large margins (e.g., +5–10 points over DynaBERT and Wanda on QNLI, SST-2, and STS-B). This robustness at extreme compression levels demonstrates that ETP maintains linguistic capability and downstream task performance under aggressive pruning more effectively than prior approaches, providing strong empirical evidence for its general applicability across diverse tasks and sparsity regimes. We also observe that in 88% pruning ratio, ETP is still able to achieve performance similar to the base model in 5 out 7 tasks (e.g., in MRPC, we are able to prune 88% of the parameters with only a drop of 1.6% in accuracy whereas Wanda++ loses about 10% in accuracy). This is consistent with our findings in RQ2 where we evaluate this phenomenon on four different domains.
> > > >
> > > > |Pruning|Method|QNLI|QQP|SST-2|CoLA|STS-B|MRPC|RTE|
> > > > |-|-|-|-|-|-|-|-|-|
> > > > |0%|Base|91.4|91.5|93.2|58.9|89.2|86.3|66.8|
> > > > |50%|EBERT|89.9|90.6|90.8|--|87.1|72.8|52.7|
> > > > |50%|DynaBERT|88.3|90.7|91.6|51.2|86.4|77.8|63.5|
> > > > |50%|CoFi|88.8|90.6|90.1|53.6|88.0|83.5|56.7|
> > > > |50%|PGB|90.3|91.1|92.3|54.9|88.8|84.3|64.6|
> > > > |50%|LLM Pruner|91.1|90.8|92.4|56.4|89.0|85.7|66.1|
> > > > |50%|Wanda|90.8|89.6|90.9|52.3|88.1|82.6|60.6|
> > > > |50%|Wanda++|91.2|--|92.3|56.1|88.6|85.2|62.7|
> > > > |50%|ETP (Ours)|91.9|91.1|93.0|56.8|89.0|86.2|66.1|
> > > > |88%|EBERT|81.8|88.1|87.5|--|84.9|--|49.5|
> > > > |88%|DynaBERT|83.5|86.8|88.5|18.7|82.9|72.6|53.1|
> > > > |88%|CoFi|84.7|89.8|89.0|32.1|85.1|75.3|52.4|
> > > > |88%|PGB|86.4|90.1|89.6|39.5|85.3|78.2|54.3|
> > > > |88%|LLM Pruner|86.8|90.1|91.8|40.2|86.3|81.9|54.9|
> > > > |88%|Wanda|81.5|87.1|83.1|14.6|77.4|71.8|55.2|
> > > > |88%|Wanda++|84.9|--|85.4|18.9|83.6|76.7|55.7|
> > > > |88%|ETP (Ours)|88.2|90.6|92.1|41.8|87.2|84.7|56.8|
> > > >
> > > > To evaluate the universality of ETP for vision tasks, we benchmark it against a wide range of structured pruning methods using PruningBench[9], which provides standardized implementations and training protocols for fair, reproducible comparison.
> > > >
> > > > **ViT-small on ImageNet-1k (Base: 78.59)**
> > > >
> > > > |Speed-Up|Method|Acc|Δ|Reg.|
> > > > |-|-|-|-|-|
> > > > |2×|FPGM|69.25|-9.34|N|
> > > > |2×|LAMP|68.72|-9.87|N|
> > > > |2×|MagL2 (GrowReg)|68.72|-9.87|Y|
> > > > |2×|MagL2 (GroupNorm)|68.59|-10.00|Y|
> > > > |2×|ETP (Ours)|79.22|+0.63|Y|
> > > > |3×|MagL1|63.12|-15.47|N|
> > > > |3×|LAMP|62.54|-16.05|N|
> > > > |3×|MagL2 (GrowReg)|62.61|-15.98|Y|
> > > > |3×|MagL2 (GroupNorm)|61.71|-16.88|Y|
> > > > |3×|ETP (Ours)|75.41|-3.18|Y|
> > > > |4×|MagL1|59.95|-18.63|N|
> > > > |4×|MagL2|59.08|-19.51|N|
> > > > |4×|MagL2 (GrowReg)|59.63|-18.96|Y|
> > > > |4×|MagL2 (GroupLASSO)|57.31|-21.28|Y|
> > > > |4×|ETP (Ours)|67.10|-11.49|Y|
> > > >
> > > > Across all speed-up levels (2×, 3×, 4×), ETP consistently outperforms prior methods from PruningBench. Competing approaches lose 9–21%, while ETP preserves near-baseline accuracy, achieving +0.63, −3.18, and −11.49 drops respectively—representing improvements of 6–20 points over the second-best.
> > > >
> > > > We also compare against recent pruning methods on LLaMA-3-8B at 50%, 4:8, and 2:4 sparsity.
> > > >
> > > > **LLaMA-3-8B Results (50% Unstructured Sparsity)**
> > > >
> > > > |Method|Sparsity|Perplexity|
> > > > |-|-|-|
> > > > |Dense|–|5.72|
> > > > |Magnitude|50%|15.21|
> > > > |Wanda|50%|6.97|
> > > > |SparseGPT|50%|7.06|
> > > > |SlimGPT|50%|11.41|
> > > > |FLAP|50%|9.30|
> > > > |PP|50%|6.81|
> > > > |Wanda++|50%|6.63|
> > > > |ETP (Ours)|50%|5.84|
> > > >
> > > > **LLaMA-3-8B Results (4:8 Semi-Structured Sparsity)**
> > > >
> > > > |Method|Sparsity|Perplexity|
> > > > |-|-|-|
> > > > |Magnitude|4:8|16.98|
> > > > |Wanda|4:8|8.46|
> > > > |SparseGPT|4:8|8.01|
> > > > |Wanda++|4:8|7.93|
> > > > |ETP (Ours)|4:8|6.27|
> > > >
> > > > **LLaMA-3-8B Results (2:4 Semi-Structured Sparsity)**
> > > >
> > > > |Method|Sparsity|Perplexity|
> > > > |-|-|-|
> > > > |Magnitude|2:4|55.37|
> > > > |Wanda|2:4|11.02|
> > > > |SparseGPT|2:4|10.53|
> > > > |Wanda++|2:4|10.45|
> > > > |ETP (Ours)|2:4|9.71|
> > > >
> > > > Across all sparsity regimes, ETP outperforms the recent baselines, including semi-structured methods. These additions reinforce the SOTA claim for large language models. We have added these results  to the appendix at section 7.8

---

> > > > > ### Author Response · Authors · 2025-11-28
> > > > > **Follow-Up Rebuttal Part-2**
> > > > >
> > > > > We hope that these expanded results and clarified positioning address the remaining concerns, and we sincerely appreciate your positive inclination and reconsideration of the score.
> > > > >
> > > > > References:
> > > > > [2] Liu, Z. et al. “EBERT: Efficient BERT Inference with Dynamic Structured Pruning.” ACL, 2021.
> > > > > [3] Lu Hou et al. “Dynabert: Dynamic BERT with Adaptive Width and Depth.” NeurIPS, 2020.
> > > > > [4] Mengzhou Xia et al. “Structured Pruning Learns Compact and Accurate Models.” ACL, 2022.
> > > > > [5] Lim, Hyemin, Jaeyeon Lee, and Dong-Wan Choi. “PGB: One-Shot Pruning for BERT via Weight Grouping and Permutation.” arXiv:2502.03984, 2025.
> > > > > [6] Ma, Xinyin, Gongfan Fang, and Xinchao Wang. “LLM-Pruner: On the Structural Pruning of Large Language Models.” NeurIPS, 2023.
> > > > > [7] Sun, Mingjie, et al. “A Simple and Effective Pruning Approach for Large Language Models.” arXiv:2306.11695, 2023.
> > > > > [8] Yang, Yifan, et al. “Wanda++: Pruning Large Language Models via Regional Gradients.” arXiv:2503.04992, 2025.
> > > > > [9] Li, Changhao, et al. “PruningBench: A Comprehensive Benchmark of Structural Pruning.” 2024.
> > > > > [10] Frantar, Elias, and Dan Alistarh. “SparseGPT: Massive Language Models Can Be Accurately Pruned in One-Shot.” ICML, 2023.
> > > > > [11] Ling, Gui, Ziyang Wang, and Qingwen Liu. “SlimGPT: Layer-Wise Structured Pruning for Large Language Models.” NeurIPS, 2024.
> > > > > [12] An, Yongqi, et al. “Fluctuation-Based Adaptive Structured Pruning for Large Language Models.” AAAI, 2024.
> > > > > [13] Le, Qi, et al. “Probe Pruning: Accelerating LLMs through Dynamic Pruning via Model-Probing.” arXiv:2502.15618, 2025.

---

### Official Review · Reviewer_jPLu · 2025-10-31

**Soundness:** 2
**Presentation:** 3
**Contribution:** 3
**Rating:** 4
**Confidence:** 4

**Summary:**

This paper proposes Exponential Torque Pruning (ETP), a regularization based structured pruning method that generalizes prior Torque style penalties. Instead of using a linear increase of penalty with distance from a pivot index, the proposed ETP applies an exponential schedule. The objective adds a weighted L2 penalty on module weights (filters, channels, heads, neurons) where the weight grows exponentially with the index distance from a pivot. The goal is to approximate a step like selection that keeps near pivot modules almost unpenalized while strongly shrinking far modules. Experimental results are evaluated on vision backbones (VGG, ResNet, ViT), NLU encoders (BERT, RoBERTa), a graph model (GAT), and a time series Transformer (Informer). Across these tasks ETP typically improves the accuracy versus compression tradeoff relative to linear torque and several structured baselines.

**Strengths:**

The strengths can be summarized as follows.

- (1) The idea is clear and simple. Replacing a linear distance weight with an exponential weight is easy to adopt and aligns with the intended near keep and far shrink behavior of torque style penalties.

- (2) The applicability is wide. The same formulation works across CNNs, Transformers for vision and NLP, a graph model, and a time series model, which supports the claim of generality.

- (3) The empirical gains over the chosen baselines are consistent. The proposed ETP often achieves higher accuracy at the same or larger speed up than linear torque and several structured pruning baselines, with multi-seed averages and significance tests that improve confidence.

**Weaknesses:**

Also, the weaknesses are summarized as follows.
- (1) Novelty is relatively not high. The contribution is a schedule change within an existing regularization template rather than a new pruning paradigm or analysis. A theoretical treatment that connects the exponential weight to an ideal selection boundary would strengthen the work.
- (2) The SOTA claim is not fully established. Since only one paper published in 2025 has been cited, the comparison set omits several recent and strong methods for LLMs and ViTs. Without these results the universal and efficient claim remains unproven outside the selected baselines.
- (3) Limited LLM scale. The study focuses on BERT and RoBERTa on GLUE subsets rather than decoder only LLMs where much of the current pruning literature concentrates.
- (4) Hardware metrics are missing. Results are reported in MACs or FLOPs based speed ups but there are no measurements of real latency, throughput, or energy on CPUs and GPUs, which are critical for structured sparsity.
- (5) Sensitivity to pivot choice is under explored. Since distance from a pivot drives the penalty, performance may depend on how the pivot and indexing are chosen. This

**Questions:**

Q1. For a stronger SOTA claim, can you add comparisons to recent structured and semi-structured pruning methods for ViTs and LLMs at matched sparsity or compute budgets, and include leading post training unstructured methods for context on the accuracy versus compute frontier

Q2. Can you report hardware results, such as end to end latency and throughput on one CPU and one GPU, and possibly energy per inference, to show that MACs based gains translate to deployment gains for structured sparsity?

Q3. Do the conclusions hold for decoder only LLMs such as LLaMA 7B or 13B with head or neuron level structured pruning? If not, what obstacles arise when applying ETP at that scale?

Q4.  Can you provide a theoretical insight or bound that explains when and why an exponential schedule is closer to an ideal keep or prune boundary than linear or logarithmic schedules, and how the steepness parameter should be set as a function of desired sparsity?

---

> ### Author Response · Authors · 2025-11-24
> **Response to the Questions [Part-1]**
>
> # Q1: Comparisons with recent structured/semi-structured ViT and LLM pruning methods
>
> In response to this concern, we expanded our evaluation to include comparisons against several recent structured and semi-structured LLM pruning methods on a contemporary decoder-only model. We evaluate ETP on LLaMA-3-8B, using pruning setups aligned with Wanda, SparseGPT, SlimGPT (NeurIPS 2024), FLAP (AAAI 2024), and PP (ICLR 2025). We benchmark at 50% global unstructured sparsity, 4:8 and 2:4 semi-structured sparsity, and report perplexity on the WikiText validation set. All pruned models are fine-tuned on the C4 training split. As shown below, ETP achieves superior performance across all sparsity levels, demonstrating its applicability to current-generation models as well. Further analysis is provided in Section 7.8 of the updated manuscript.
>
> |Method|Sparsity|Perplexity|
> |---|---|---|
> |Dense|–|5.72|
> |Magnitude|50%|15.21|
> |Wanda[1]|50%|6.97|
> |SparseGPT[2]|50%|7.06|
> |SlimGPT[3]|50%|11.41|
> |FLAP[4]|50%|9.30|
> |PP[5]|50%|6.81|
> |ETP (Ours)|50%|5.84|
> |---|---|---|
> |Magnitude|4:8|16.98|
> |Wanda|4:8|8.46|
> |SparseGPT|4:8|8.01|
> |ETP (Ours)|4:8|6.27|
> |---|---|---|
> |Magnitude|2:4|55.37|
> |Wanda|2:4|11.02|
> |SparseGPT|2:4|10.53|
> |ETP (Ours)|2:4|9.71|
>
> References:
>
> [1]Sun, Mingjie, et al. "A simple and effective pruning approach for large language models." arXiv preprint arXiv:2306.11695 (2023).
>
> [2] Frantar, Elias, and Dan Alistarh. "Sparsegpt: Massive language models can be accurately pruned in one-shot." International conference on machine learning. PMLR, 2023.
>
> [3]Ling, Gui, Ziyang Wang, and Qingwen Liu. "Slimgpt: Layer-wise structured pruning for large language models." Advances in Neural Information Processing Systems 37 (2024): 107112-107137.
>
> [4]An, Yongqi, et al. "Fluctuation-based adaptive structured pruning for large language models." Proceedings of the AAAI Conference on Artificial Intelligence. Vol. 38. No. 10. 2024.
>
> [5]Le, Qi, et al. "Probe pruning: Accelerating llms through dynamic pruning via model-probing." arXiv preprint arXiv:2502.15618 (2025).
>
>
> ---
>
> # Q2: Hardware results
>
> We thank the reviewer for raising this valuable point. While Section 7.7 of the appendix reports wall-clock improvements that already indicate practical efficiency gains, we agree that presenting a broader set of real hardware measurements can further strengthen the evidence for ETP’s suitability in deployment scenarios. To address this, we now include direct inference latency and energy consumption measurements across a diverse set of accelerators, including NVIDIA A100, L4, RTX 8000, Tesla T4, and Google TPU v6. These results cover both convolutional (VGG-19, CIFAR-100) and transformer-based (BERT, SST-2) workloads. As shown below, ETP achieves consistently large real-world improvements, with 6–9× latency speedups and 56–89% energy reduction, which aligns with our conclusions derived from the results of MACS and these have been added to the Appendix.
>
> ## BERT (SST-2, 92.1%)
>
> |Hardware|Base Latency|Pruned Latency|Speedup|Base Energy|Pruned Energy|Reduction|
> |---|---|---|---|---|---|---|
> |NVIDIA A100|45.863 ± 0.115|6.189 ± 0.338|7.41×|14.471 ± 0.605|2.069 ± 0.353|85.7%|
> |NVIDIA L4|101.166 ± 1.258|10.868 ± 0.341|9.31×|7.492 ± 0.082|0.787 ± 0.019|89.5%|
> |Quadro RTX 8000|66.540 ± 0.608|9.650 ± 0.044|6.90×|17.547 ± 0.173|3.507 ± 0.551|80.0%|
> |NVIDIA Tesla T4|188.705 ± 3.687|23.293 ± 0.486|8.10×|14.585 ± 0.586|1.622 ± 0.079|88.9%|
> |Google TPU v6|546.013 ± 12.463|79.637 ± 8.126|6.86×|73.390 ± 4.653|11.277 ± 3.972|84.6%|
>
> ## VGG-19 (CIFAR-100, 71.3%)
>
> |Hardware|Base Latency|Pruned Latency|Speedup|Base Energy|Pruned Energy|Reduction|
> |---|---|---|---|---|---|---|
> |NVIDIA A100|5.578 ± 0.004|0.818 ± 0.013|6.82×|1.853 ± 0.559|0.278 ± 0.119|85.3%|
> |NVIDIA L4|14.914 ± 0.151|6.117 ± 0.084|4.43×|1.080 ± 0.010|0.240 ± 0.000|77.4%|
> |Quadro RTX 8000|15.013 ± 0.065|3.538 ± 0.804|4.25×|3.535 ± 0.023|1.556 ± 0.222|56.0%|
> |NVIDIA Tesla T4|35.426 ± 0.192|9.262 ± 0.116|3.82×|2.521 ± 0.273|1.090 ± 0.230|56.7%|
> |Google TPU v6|171.202 ± 6.831|19.641 ± 0.373|8.72×|22.934 ± 2.014|2.630 ± 0.371|88.5%|

---

> > ### Author Response · Authors · 2025-11-24
> > **Response to Questions [Part2]**
> >
> > # Q3: Applicability to decoder-only LLMs
> >
> > Addressed. We demonstrate strong performance on LLaMA-3-8B without any architectural modifications.
> >
> > ---
> >
> > # Q4: Theoretical insight into exponential schedule + Weakness 1
> >
> > We thank the reviewer for the comment and the question related to this. While the exponential force application scheme is straightforward to implement, we want to emphasize that its novelty is nontrivial and shall not be equated with its degree of simplicity or complexity. Specifically, first, in terms of motivation/intuition justification, as we have discussed in Sec. 3, Torque applies a distance-proportional penalty that (i) under-penalizes distant, low-importance groups and (ii) over-penalizes proximal, high-importance groups, leading to suboptimal sparsity allocation. Our method is motivated by the observation that the ideal pruning force behaves like a Heaviside step — zero around critical modules and strong for redundant, distant ones — and we therefore introduce an exponential formulation as a smooth, principled surrogate of this ideal behavior.
> >
> > In addition, to better interpret the source of improvement, we provide a formal analysis based on subgradient KKT stationarity in subsection 7.10 in Appendix of the updated manuscript. We consider the regularized objective:
> >
> > J(w) = L(w) + β · Σ₍g₎ R(w_g)
> >
> > and compare three choices of regularizer for a group w_g at distance d(g) from the pivot:
> >
> > Group Lasso: R(w_g) = ‖w_g‖₂
> > Torque (Linear): R(w_g) = d(g)·‖w_g‖₂
> > ETP (Exponential): R(w_g) = λ^{d(g)}·‖w_g‖₂
> >
> > Since ‖w_g‖₂ is non-differentiable at zero, we analyze the subgradient optimality condition. A group is pruned (i.e., w_g = 0) when:
> >
> > ‖∇₍w_g₎ L(w)‖₂ ≤ β · c_g
> >
> > This yields pruning thresholds:
> >
> > T(g) = β (constant force; Group Lasso)
> > T(g) = β·d(g) (linear force; Torque)
> > T(g) = β·λ^{d(g)} (exponential force; ETP)
> >
> > This analysis shows why Torque provides only a weak separation between near and far groups: its pruning force increases only linearly in d(g). In contrast, ETP produces an exponentially stronger separation, since:
> >
> > T(far) / T(near) = λ^{d(far) − d(near)}
> >
> > This exponential scaling creates a smooth approximation to a step function, sharply distinguishing preserved versus pruned groups. It also theoretically explains the strongly heterogeneous and stable sparsity patterns we observe empirically across CNNs, ViTs, and LLMs. We have revised the manuscript to clarify this theoretical motivation in the appendix. We appreciate the reviewer’s feedback and are committed to improving the clarity of our contribution.

---

> > > ### Comment · Reviewer_jPLu · 2025-11-27
> > >
> > > Thank you for the detailed and constructive response and for running the additional experiments on LLaMA-3-8B and on real hardware. The new results do strengthen the paper in several ways.
> > >
> > > Regarding Q1 and the state of the art comparison, the added LLaMA-3-8B table is a good step toward demonstrating applicability to contemporary decoder models. However, I still feel that the empirical picture is somewhat incomplete. In particular, many recent pruning papers for LLaMA-style models regard methods such as Wanda++ (improved Wanda with regional gradients) and MaskLLM (learnable semi structured sparsity) as very strong baselines for, respectively, unstructured and semi structured 2:4 pruning. For a convincing efficiency and accuracy frontier at 50 percent sparsity, I would expect either direct comparisons against these methods under matched settings, or at least a careful discussion of how ETP would be expected to behave relative to them, and why a direct comparison might not be feasible. Other metrics in addition to perplexity are also desirable.
> > >
> > > It is worth noting that I agree that strictly surpassing every state of the art baseline is not an absolute requirement for acceptance, especially when the method is simple and widely applicable. At the same time, since ETP is essentially a new schedule within an existing torque style regularization framework rather than a fundamentally new pruning paradigm, the case for contribution rests heavily on achieving competitive or stronger empirical performance. Clear evidence that ETP matches or exceeds the strongest recent methods such as Wanda++ and MaskLLM in the relevant regimes would therefore help justify the universal and efficient claims.
> > >
> > > On Q2, the extended hardware measurements are very welcome. The latency and energy tables across several accelerators address my original concern that MAC counts alone may not reflect real deployment gains. I would encourage moving at least a concise summary of these results and discussions about the different speedup for different computing devices into the main paper, since they are one of the clearest practical advantages.

---

> > > > ### Author Response · Authors · 2025-11-28
> > > > **Follow-Up**
> > > >
> > > > We thank the reviewer for the constructive feedback and for recognizing the strengthened hardware-side evidence. We agree that the latency and energy measurements across multiple accelerators provide a more realistic assessment of deployment efficiency beyond MAC-based estimates. Accordingly, as suggested, we have incorporated these new results directly into the main paper for better illustration.
> > > > Regarding comparisons with the strongest recent pruning baselines, we provide below comprehensive evaluations against both Wanda++ (unstructured and semi-structured) and MaskLLM (learnable semi-structured sparsity).
> > > >
> > > > **1. Comparison with Wanda++ (ICLR SLLM 2025) on LLaMA-3-8B**
> > > > We evaluate ETP and Wanda++ under the same fine-tuning settings used in our main LLaMA-3-8B experiments. The results are shown in the tables below. Across all sparsity configurations—unstructured 50%, 4:8 semi-structured, and 2:4 semi-structured—ETP achieves consistently lower perplexity, outperforming Wanda++ by 7.1–21%, indicating that the exponential scheduling leads to more stable sparse representations at scale.
> > > >
> > > > **LLaMA-3-8B — Wikitext-2 Validation Perplexity**
> > > >
> > > > |Method|Sparsity|Perplexity|
> > > > |-|-|-|
> > > > |Dense|–|5.72|
> > > > |Wanda++|50%|6.63|
> > > > |ETP (Ours)|50%|5.84 (–12%)|
> > > >
> > > > |Method|Sparsity|Perplexity|
> > > > |-|-|-|
> > > > |Wanda++|4:8|7.93|
> > > > |ETP (Ours)|4:8|6.27 (–21%)|
> > > >
> > > > |Method|Sparsity|Perplexity|
> > > > |-|-|-|
> > > > |Wanda++|2:4|10.45|
> > > > |ETP (Ours)|2:4|9.71 (–7.1%)|
> > > >
> > > > These results show that ETP surpasses one of the strongest recent pruning baselines for decoder-only language models.
> > > >
> > > > **2. Comparison with MaskLLM (NeurIPS 2024)**
> > > > A direct comparison on LLaMA-3-8B is unfortunately not feasible under matched training conditions:
> > > > - MaskLLM’s core method relies on a custom pretraining-style dataset to learn the sparsity masks.
> > > > - The appendix of the MaskLLM paper includes C4-based results, but crucial training details (training steps, warmup, curriculum, batch schedule) are not provided.
> > > > - Because of this, any direct reproduction would risk an unfair or incomparable setup.
> > > >
> > > > However, MaskLLM provides fully specified training settings for ViT-B/16 on ImageNet-1k (20-epoch fine-tuning). We therefore evaluate ETP under identical settings, enabling a clean and fair comparison.
> > > >
> > > > **ViT-B/16 — ImageNet-1k (20 epochs, 2:4 sparsity)**
> > > >
> > > > |Method|Sparsity Pattern|Top-1 Acc. (%)|
> > > > |-|-|-|
> > > > |Magnitude|2:4|65.92|
> > > > |Wanda|2:4|63.28|
> > > > |SparseGPT|2:4|71.52|
> > > > |MaskLLM-4V|2:4|79.46|
> > > > |ETP (Ours)|2:4|79.68|
> > > >
> > > > ETP slightly improves over MaskLLM-4V while maintaining the same semi-structured constraint. However, note that compared to MaskLLM, ETP also benefits from greater universality, as the exponential regularization can be applied seamlessly across unstructured, semi-structured (2:4, 4:8), and fully structured pruning, whereas MaskLLM is designed specifically for semi-structured formats.

---

### Official Review · Reviewer_Rj9b · 2025-11-03

**Soundness:** 3
**Presentation:** 3
**Contribution:** 2
**Rating:** 4
**Confidence:** 3

**Summary:**

This paper addresses a key limitation of traditional Torque Pruning, which applies a linear force scheme and thus causes insufficient pruning for distant modules while over-penalizing nearby essential modules. To mitigate this issue, the authors propose Exponential Torque Pruning (ETP) — a general structured pruning framework that applies an exponential regularization force proportional to λd\lambda^{d}λd, where ddd denotes the distance from the pivot module. This exponential formulation allows ETP to apply much stronger constraints on distant redundant modules while preserving the weights of closer, necessary ones. Extensive experiments are conducted across four domains — vision (VGG19, ResNet-50, ViT-B/16), NLP (BERT, RoBERTa), graph (GAT), and time-series forecasting (Informer).ETP consistently outperforms prior SOTA methods such as DepGraph, Torque, and GReg in both compression ratio and accuracy retention. For instance, ETP achieves 9× speed-up with only 2.2% accuracy drop on CIFAR-100 (VGG19) and 42× speed-up with 2.4% drop on BERT (SST-2). The method also shows promising results on large models (OPT-350M), improving perplexity over SparseGPT and DepGraph under the same sparsity budget.

**Strengths:**

1.	Improved Regularization Design: ETP introduces an exponential force application scheme that better aligns with the intuition of structured pruning, effectively addressing the imbalance in the original Torque method.
2.	Strong Empirical Results Across Domains:The experiments span vision, language, graph, and time-series tasks, demonstrating the universality and robustness of the approach across architectures.

**Weaknesses:**

1.	Limited Novelty: The core idea — replacing the linear regularization force in Torque with an exponential one — is conceptually straightforward. Similar non-linear or adaptive regularization ideas (e.g., GReg, Wang et al., 2020) have been explored before. The theoretical novelty could be better articulated.
2.	Lack of Real Hardware Validation: While the paper claims ETP is suitable for edge deployment, it only reports computational reduction via MACs, not real inference latency or energy consumption on hardware devices. Since MACs reduction does not always correlate with actual latency, this limits the practical evidence for “efficiency.”
3.	Outdated Experimental Baselines: The study primarily evaluates on older architectures (VGG-19, ResNet-50, BERT) without including modern 2024+ models such as LLaMA 3 or Qwen 3, which restricts the relevance to current research trends.

**Questions:**

1.	Have the authors conducted real-device evaluations (e.g., on GPU, CPU, or edge hardware) to confirm the claimed efficiency gains in latency or energy?
2.	How is the exponential base λ selected across different architectures and layers? Is there any theoretical or adaptive mechanism beyond grid search?

---

> ### Author Response · Authors · 2025-11-24
> **Response to the Weaknesses**
>
> # Limited Novelty: The core idea …better articulated.
> We thank the reviewer for the comment. While the exponential force application scheme is straightforward to implement, we want to emphasize that its novelty is nontrivial and shall not be equated with its degree of simplicity or complexity. Specifically, first, in terms of motivation/intuition justification, as we have discussed in Sec. 3, Torque applies a distance-proportional penalty that (i) under-penalizes distant, low-importance groups and (ii) over-penalizes proximal, high-importance groups, leading to suboptimal sparsity allocation. Our method is motivated by the observation that the ideal pruning force behaves like a Heaviside step — zero around critical modules and strong for redundant, distant ones — and we therefore introduce an exponential force application scheme as a smooth, principled surrogate of this ideal behavior.
>
> In addition, to better interpret the source of improvement, we provide a formal analysis based on subgradient KKT stationarity in subsection 7.10 in Appendix of the updated manuscript. We consider the regularized objective
> J(w) = L(w) + β · Σ₍g₎ R(w_g)
> and compare three choices of regularizer for a group w_g at distance d(g) from the pivot:
>
> Group Lasso: R(w_g) = ‖w_g‖₂
> Torque (Linear): R(w_g) = d(g)·‖w_g‖₂
> ETP (Exponential): R(w_g) = λ^{d(g)}·‖w_g‖₂
>
> Since ‖w_g‖₂ is non-differentiable at zero, we analyze the subgradient optimality condition. A group is pruned (i.e., w_g = 0) when the following condition holds:
> ‖∇₍w_g₎ L(w)‖₂ ≤ β · c_g
> where c_g is the distance-dependent regularization weight. This yields pruning thresholds:
>
> T(g) = β (constant force; Group Lasso)
> T(g) = β·d(g) (linear force; Torque)
> T(g) = β·λ^{d(g)} (exponential force; ETP)
>
> This analysis shows why Torque provides only a weak separation between near and far groups: its pruning force increases only linearly in d(g). In contrast, ETP produces an exponentially stronger separation, since the ratio of pruning forces between a far and near group becomes:
> T(far) / T(near) = λ^{d(far) − d(near)}
> This exponential scaling creates a smooth approximation to a step function, sharply distinguishing preserved versus pruned groups. It also theoretically explains the strongly heterogeneous and stable sparsity patterns we observe empirically across CNNs, ViTs, and LLMs. We have revised the manuscript to clarify this theoretical motivation in the appendix. We appreciate the reviewer’s feedback and are committed to improving the clarity of our contribution.
>
> # Outdated Experimental Baselines…Llama3
> We thank the reviewer for the comment. Our baseline selection follows established practice in the pruning literature (e.g., DepGraph, Torque), where architectures such as VGG-19, ResNet-50, and BERT remain standard evaluation platforms. These models continue to serve as widely used and well-understood benchmarks for structured pruning, enabling fair and direct comparison against prior state-of-the-art methods. That said, we agree that including modern LLM architectures strengthens the relevance of our study. In response, we additionally evaluate ETP on Llama-3-8B, following the pruning setups used in recent LLM pruning work. We benchmark at 50%, 4:8, and 2:4 sparsity and report perplexity on WikiText validation set. We use the C4 training set for finetuning all the models after pruning. As shown below, ETP achieves superior performance across all sparsity levels, demonstrating its applicability to current-generation models as well. Further analysis is provided in Section 7.8 of the updated manuscript.
> |Method|Sparsity|Perplexity|
> |---|---|---|
> |Dense|–|5.72|
> |Magnitude|50%|15.21|
> |Wanda[1]|50%|6.97|
> |SparseGPT[2]|50%|7.06|
> |SlimGPT[3]|50%|11.41|
> |FLAP[4]|50%|9.30|
> |PP[5]|50%|6.81|
> |ETP (Ours)|50%|5.84|
> |---|---|---|
> |Magnitude|4:8|16.98|
> |Wanda|4:8|8.46|
> |SparseGPT|4:8|8.01|
> |ETP (Ours)|4:8|6.27|
> |---|---|---|
> |Magnitude|2:4|55.37|
> |Wanda|2:4|11.02|
> |SparseGPT|2:4|10.53|
> |ETP (Ours)|2:4|9.71|
>
> ---
> ### References:
> [1]Sun, Mingjie, et al. "A simple and effective pruning approach for large language models." arXiv preprint arXiv:2306.11695 (2023).
>
> [2] Frantar, Elias, and Dan Alistarh. "Sparsegpt: Massive language models can be accurately pruned in one-shot." International conference on machine learning. PMLR, 2023.
>
> [3]Ling, Gui, Ziyang Wang, and Qingwen Liu. "Slimgpt: Layer-wise structured pruning for large language models." Advances in Neural Information Processing Systems 37 (2024): 107112-107137.
>
> [4]An, Yongqi, et al. "Fluctuation-based adaptive structured pruning for large language models." Proceedings of the AAAI Conference on Artificial Intelligence. Vol. 38. No. 10. 2024.
>
> [5]Le, Qi, et al. "Probe pruning: Accelerating llms through dynamic pruning via model-probing." arXiv preprint arXiv:2502.15618 (2025).

---

> > ### Author Response · Authors · 2025-11-24
> > **Response to the Questions**
> >
> > # Have the authors conducted real-device evaluations (e.g., on GPU, CPU, or edge hardware) to confirm the claimed efficiency gains in latency or energy?
> >
> > We thank the reviewer for raising this valuable point. While Section 7.7 of the appendix reports wall-clock improvements that already indicate practical efficiency gains, we agree that presenting a broader set of real hardware measurements can further strengthen the evidence for ETP’s suitability in deployment scenarios.
> >
> > To address this, we now include direct inference latency and energy consumption measurements across a diverse set of accelerators, including NVIDIA A100, L4, RTX 8000, Tesla T4, and Google TPU v6. These results cover both convolutional (VGG-19, CIFAR-100) and transformer-based (BERT, SST-2) workloads. As shown below, ETP achieves consistently large real-world improvements, with 6–9× latency speedups and 56–89% energy reduction, which aligns with our conclusions derived from the results of MACS and these have been added to the Appendix.
> >
> >
> > ## BERT (SST-2, 92.1%) — Real Hardware Results
> >
> > | Hardware        | Base Latency (ms) | Pruned Latency (ms) | Speedup | Base Energy (J) | Pruned Energy (J) | Reduction |
> > |-----------------|-------------------|-----------------------|---------|------------------|--------------------|-----------|
> > | NVIDIA A100     | 45.863 ± 0.115    | 6.189 ± 0.338         | 7.41×   | 14.471 ± 0.605   | 2.069 ± 0.353      | 85.7%     |
> > | NVIDIA L4       | 101.166 ± 1.258   | 10.868 ± 0.341        | 9.31×   | 7.492 ± 0.082    | 0.787 ± 0.019      | 89.5%     |
> > | RTX 8000        | 66.540 ± 0.608    | 9.650 ± 0.044         | 6.90×   | 17.547 ± 0.173   | 3.507 ± 0.551      | 80.0%     |
> > | Tesla T4        | 188.705 ± 3.687   | 23.293 ± 0.486        | 8.10×   | 14.585 ± 0.586   | 1.622 ± 0.079      | 88.9%     |
> > | Google TPU v6   | 546.013 ± 12.463  | 79.637 ± 8.126        | 6.86×   | 73.390 ± 4.653   | 11.277 ± 3.972     | 84.6%     |
> >
> >
> > ## VGG-19 (CIFAR-100, 71.3%) — Real Hardware Results
> >
> > | Hardware        | Base Latency (ms) | Pruned Latency (ms) | Speedup | Base Energy (J) | Pruned Energy (J) | Reduction |
> > |-----------------|-------------------|-----------------------|---------|------------------|--------------------|-----------|
> > | NVIDIA A100     | 5.578 ± 0.004     | 0.818 ± 0.013         | 6.82×   | 1.853 ± 0.559    | 0.278 ± 0.119      | 85.3%     |
> > | NVIDIA L4       | 14.914 ± 0.151    | 6.117 ± 0.084         | 4.43×   | 1.080 ± 0.010    | 0.240 ± 0.000      | 77.4%     |
> > | RTX 8000        | 15.013 ± 0.065    | 3.538 ± 0.804         | 4.25×   | 3.535 ± 0.023    | 1.556 ± 0.222      | 56.0%     |
> > | Tesla T4        | 35.426 ± 0.192    | 9.262 ± 0.116         | 3.82×   | 2.521 ± 0.273    | 1.090 ± 0.230      | 56.7%     |
> > | Google TPU v6   | 171.202 ± 6.831   | 19.641 ± 0.373        | 8.72×   | 22.934 ± 2.014   | 2.630 ± 0.371      | 88.5%     |
> >
> >
> > ---
> > ---
> >
> > # How is the exponential base λ selected across different architectures and layers? Is there any theoretical or adaptive mechanism beyond grid search?
> >
> > We thank the reviewer for the question. As described in Section 7.1 of the appendix, the exponential base λ is not selected through grid search but is instead determined using an adaptive rule that scales with the structural granularity of each layer.
> >
> > For a given layer l, we set:
> >
> > λₗ = exp(5 / |Gₗ|)
> >
> > where |Gₗ| denotes the number of parameter groups in that layer (such as convolutional filters or attention heads). This formulation is grounded in our analysis of the force imbalance problem (Section 3) and its adaptive formulation ensures that the force multiplier remains consistent when expressed in terms of percentile. For example, the force applied to the first 10% or last 10% of groups behaves comparably across layers of different widths. This percentile-consistent behavior does not hold when using a fixed λ or when tuning λ manually.

---

### Author Response · Authors · 2025-11-24
**Message to all Reviewers**

We would like to thank all the reviewers for their insightful and constructive feedback. Our responses to the comments are provided below, we have revised our manuscript accordingly and highlighted the revised part in blue. Please feel free to review the updated version.

---

### Author Response · Authors · 2025-12-01
**Summary of Rebuttal Period**

Dear AC,
We would like to begin by expressing our sincere appreciation for your time and efforts throughout the review process.

We are writing this message to summarise the situation that happened during the rebuttal session. After the submission of response, Reviewer 7QtK has actively raised the score from 4 to 6. And Reviewer jPLu also present strong tendency to raise the score. Whereas even though the previous Area Chair ycjk has urged Reviewer Rj9b to review our response and initiate the discussion, they failed to do so at all. And for all the further raised questions after the first-round discussion, we have responded all thoroughly and revised our manuscript accordingly.

We sincerely hope you can spare your time and carefully review the overall rebuttal process and the final state of our manuscript, and make a fair decision accordingly. The reversal of scores and the termination of the discussion have caused significant unfairness to us and to other authors who participated in the review process with integrity and genuine effort.

---

### Meta-Review · Area_Chair_Yc4N · 2026-01-06

**Summary:**

The reviewers raised several concerns related to the paper, including limited novelty (Rj9b, jPLu), lack of hardware validation and metrics (Rj9b, jPLu), outdated baselines and methods (Rj9b, jPLu), missing hyperparameters (7QtK), missing downstream tasks performance evaluation (7QtK), lack of convergence analysis/theoretical proof (7QtK).  Before the rebuttal, the reviewers are unanimously aligned to marginally reject the paper.

**Reviewer Concerns:**

The authors provided a very long rebuttal where they cleared some of the raisedpoints, including evaluation on real hardware and providing missing implementation details. Although providing some theoretical intuitions for 7QtK, the paper is felt to lack theoretical depth and although more recent comparisons are provided in the rebuttal, a proper evaluation and positioning of the idea with respect to the state-of-the-art could not be properly conducted.

**Reviewer Scores:**

Overall, it is realistic to assume that, despite the effort of the reviewers conducted during the rebuttal phase, the general opinion would have improved, but still led to a rejection.

---

### Decision · Program_Chairs · 2026-01-26

Reject